# Probing Visual Language Priors in VLMs

**Tiange Luo** [1] [*]   **Ang Cao** [1] [*]   **Gunhee Lee** [2]   **Justin Johnson** [1] [†]   **Honglak Lee** [1] [2] [†]

## Abstract

Vision-Language Models (VLMs) may over-rely on visual language priors from their training data rather than true visual reasoning. To investigate this, we introduce ViLP, a benchmark featuring deliberately out-of-distribution images synthesized via image generation models and out-of-distribution Q&A pairs. Each question in ViLP is coupled with three potential answers and three corresponding images: one that can be resolved by text priors alone and two that demand visual reasoning. Although humans achieve near-perfect accuracy, modern VLMs falter; for instance, GPT-4o achieves only 66.17% on ViLP. To alleviate this, we propose a self-improving framework in which models generate new VQA data and then apply pixel-level and semantic corruptions to form "good-bad" image pairs for self-training. Our proposed training objective, Image-DPO, compels VLMs to focus more on the actual visual inputs, and we demonstrate its effectiveness in LLaVA-v1.5 and Cambrian. Project Page: ViLP.

## 1. Introduction

Vision-Language Models (VLMs) have advanced text-image interaction, bridging the gap between visual and textual data (Achiam et al., 2023; Team et al., 2023). However, a persistent challenge for learning-based models, such as ResNets and CLIPs, lies in their reliance on learned priors from the training data, sometimes overlooking visual cues when answering questions (Agrawal et al., 2016; Prabhu et al., 2023). For example, when shown a torus-shaped soccer ball (Figure 1), a model might incorrectly identify it as a sphere due to strong language priors. Simultaneously, these models may adhere to learned visual priors (Thrush et al., 2022; Sterz et al., 2024), which hampers their ability to comprehend out-of-distribution visual cues, such as a zebra

with atypical spot patterns (Figure 1), which humans would easily discern. This raises an important question: do today's VLMs still over-rely on learned visual language priors, especially given that they rely on far fewer image-text pairs than the internet-scale text corpora used for pretraining?

To investigate this, we probe the **Vi**sual **L**anguage **P**riors of VLMs by constructing Question-Image-Answer (QIA) triplets that deliberately deviate from the training data distribution. Unlike existing benchmarks that typically rely on internet-sourced images (Goyal et al., 2017; Tong et al., 2024), which inadvertently favor the visual language priors embedded in VLM training data, we utilize modern image generation models, including DALL·E-3 (Ramesh et al., 2021) and Flux, to synthesize out-of-distribution image-answer pairs that also feature notable variation in texture, shape, conceptual combinations, hallucinated elements, and proverb-based contexts.

Our benchmark, ViLP, contains 300 carefully designed questions, each paired with three distinct answers: a *Prior Answer* and two *Test Answers*, resulting in a total of 900 QIA triplets. To further challenge the priors of VLMs, we amplify language priors in questions by introducing distractor facts: each question is structured to present a distractor fact followed by a question. The distractor fact directly leads to the Prior Answer. In contrast, the two Test Answers are crafted to challenge the priors by requiring both textual and visual cues for accurate reasoning. While human participants achieved ∼98% accuracy easily, current VLMs exhibit considerable difficulty, as evidenced by a significant performance drop on our benchmarks, with GPT-4o (OpenAI, 2024) scoring only 66.17%.

Motivated by the results of ViLP, we propose Image-DPO, a self-improving method for enhancing VLM visual reasoning by increasing reliance on visual inputs. Our method employs self-generated VQAs using image generation and editing models (Podell et al., 2023; Ren et al., 2024; Brooks et al., 2023a) and applies controlled corruptions to create "good-bad" image pairs for DPO-like training (Rafailov et al., 2024). Experiments with open-source VLMs, including LLaVA-v1.5 (Liu et al., 2024a) and Cambrian (Tong et al., 2024), demonstrate its effectiveness. Moreover, we theoretically prove our objective optimizes an upper bound on the RL fine-tuning objective (Rafailov et al., 2024).

---

[*]Joint fist authorship, [†]Equal advising [1]University of Michigan [2]LGAI Research. Correspondence to: Tiange Luo <tiangel@umich.edu>, Ang Cao <ancao@umich.edu>.

*Proceedings of the 42nd International Conference on Machine Learning*, Vancouver, Canada. PMLR 267, 2025. Copyright 2025 by the author(s).

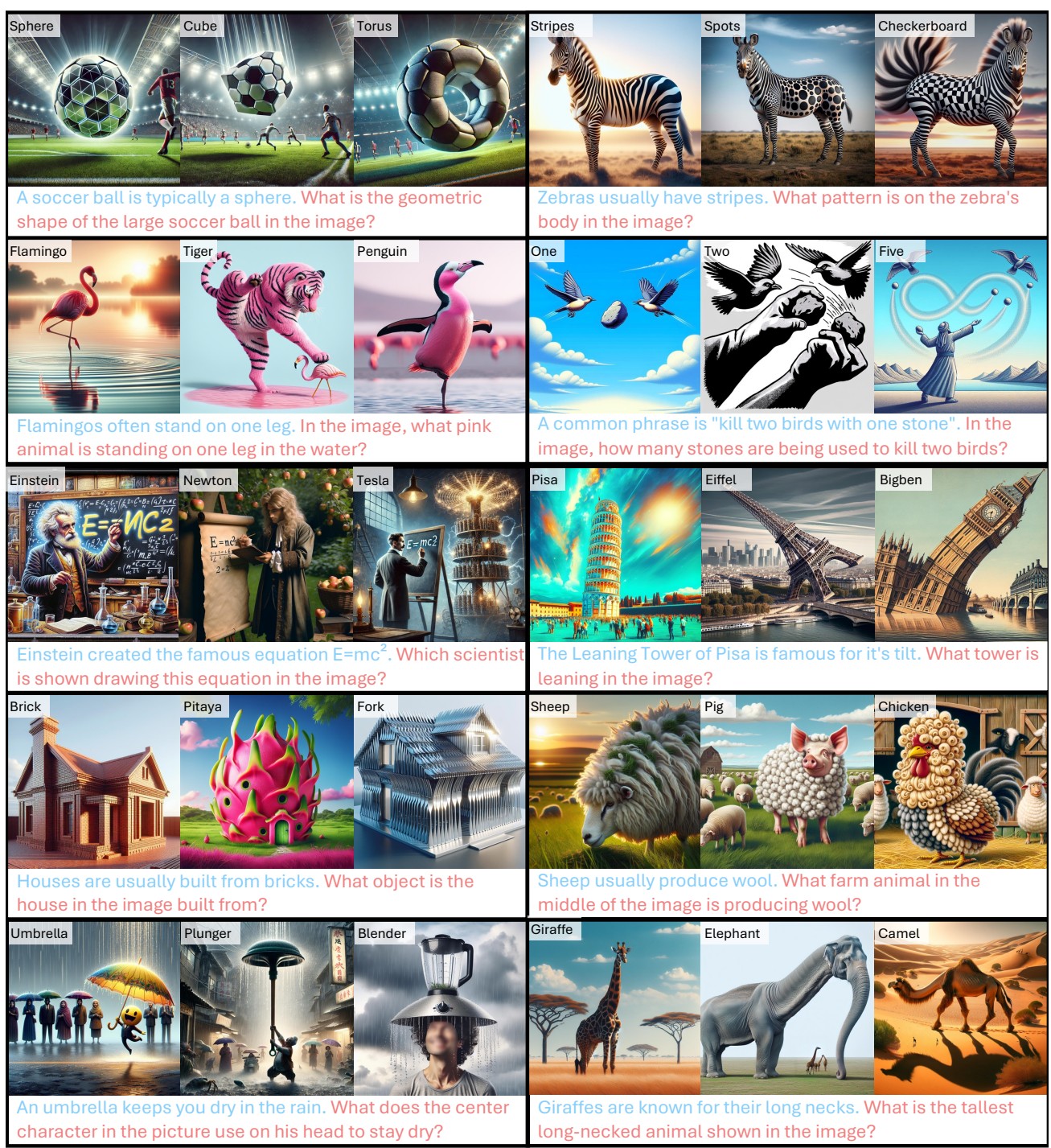

Figure 1: **Sample data from ViLP.** For the same question, ViLP provides three distinct images and corresponding answers (upper-left corner). All questions follow a consistent structure, combining a distractor fact with a question. The *Prior Answer* (first column) can be directly inferred from the question, while *Test Answers* (second & third column) rely on visual cues. Our answers are designed to be single words, and both the model and human evaluators are tasked with open-domain answering, rather than selecting from predefined options. To support this, we have developed a robust synonym and plural detection pipeline, ensuring that open-ended responses do not hinder the evaluation process. This approach also enables evaluation without relying on LLMs. Please refer to Appendix B.1 for more data samples from ViLP. We investigate the impact of image styles in Appendix B.3, where we generate more realistic images using 4o image generation. Furthermore, we include both qualitative and quantitative comparison results with Winoground (Thrush et al., 2022), Whoops!(Bitton-Guetta et al., 2023a), and HallusionBench(Guan et al., 2023) in Appendix B.4.

## 2. Related Work

**VQA Dataset:** Significant efforts have produced VQA datasets from various angles, including general VQA (Agrawal et al., 2015; Gurari et al., 2018; Fu et al., 2023; Liu et al., 2023d; Li et al., 2023a; Yu et al., 2023b; Liu et al., 2024a), reading text or charts (Singh et al., 2019a; Mathew et al., 2020; 2021; Masry et al., 2022), complex reasoning (Lu et al., 2022; 2023), composition probing (Hudson & Manning, 2019; Ma et al., 2022; Thrush et al., 2022; Hsieh et al., 2023; Li et al., 2024a), hallucinations (Rohrbach et al., 2018; Li et al., 2023c; Guan et al., 2023), common-sense reasoning (Bitton-Guetta et al., 2023b;a), and more (Majumdar et al., 2024; Sterz et al., 2024). We propose a benchmark that tests VLMs' visual reasoning when questions, answers, and images defy common patterns. Following the balanced dataset design of (Goyal et al., 2017), each question is accompanied by three answers: one aligns with language priors, and two deviate, prompting reliance on visual cues. By leveraging state-of-the-art image generation models, our benchmark challenges these priors more effectively than previous datasets built from internet images (Goyal et al., 2017; Tong et al., 2024). Furthermore, unlike the "trick" category in (Sterz et al., 2024), we first generate question-answer pairs before synthesizing images under specified constraints, creating more difficult visual reasoning tasks. Comprehensive comparisons with existing datasets appear in Appendix B.4.

**Vision Language Models and Language Priors**: Multimodal reasoning is crucial for machine intelligence, with VLMs integrating visual perception, text reasoning, instruction following, and generation for complex tasks (Tan & Bansal, 2019; Li et al., 2019; Kim et al., 2021; Wang et al., 2021b;a; Alayrac et al., 2022; Li et al., 2023b; Chen et al., 2022; Jia et al., 2021; Shen et al., 2021; Singh et al., 2021; Liu et al., 2023c;a; Zhao et al., 2023a; Chen et al., 2023; Zhu et al., 2024b; Li et al., 2024c; Dai et al., 2023; Li et al., 2024c; Yu et al., 2024; Dai et al., 2024; Deitke et al., 2024). Inspired by the success of large language models (Brown et al., 2020; OpenAI, 2023a; Touvron et al., 2023a;b; Chiang et al., 2023) and pre-trained visual encoders (Radford et al., 2021; Desai & Johnson, 2020; Caron et al., 2021; Chen et al., 2024), many recent methods leverage relatively small vision-language paired datasets (Liu et al., 2024a; Tong et al., 2024) to fine-tune connectors between LLMs and visual backbones (Liu et al., 2024a). However, these datasets are far smaller than the vast text corpora for LLM pre-training (OpenAI, 2023b; Soldaini et al., 2024), and freezing the visual encoder and LLM parameters often preserves language biases, causing visual inputs to be overshadowed (Thrush et al., 2022; Sterz et al., 2024). This challenge is amplified by deliberately generated images that expose such biases, as shown in our study. Previous works and datasets (Goyal et al., 2016; Agrawal et al., 2017; Dancette et al., 2021; Wu et al., 2022; Ramakrishnan et al., 2018; Gouthaman & Mittal, 2020) addressed these issues with curated simulators (Johnson et al., 2016) or internet imagery (Zhang et al., 2016). In this paper, we present a novel VQA benchmark featuring carefully designed questions, fact-based distractors, rare-distribution answers, and image generation techniques to produce realistic visuals that challenge learned visual language priors (Figure 1).

**Self-Rewarding VLM**: Self-rewarding LLM (Yuan et al., 2024) has shown that LLMs can generate and improving themselves in the process via Directed Peference Optimization (DPO) (Rafailov et al., 2024). This approach extends to VLMs by generating new answers for DPO training (Zhou et al., 2024a; Deng et al., 2024; Zhou et al., 2024b; Wang et al., 2024c;a; Yue et al., 2024; Liu et al., 2024b; Xiao et al., 2024). Our work aligns with these self-rewarding VLMs but differs in two key ways: (1) our proposed Image-DPO generates multiple images for a single question-image pair (rather than multiple answers); (2) rather than relying solely on existing images (Zhu et al., 2024a), Image-DPO creates diverse new images using pre-trained generative models (SDXL (Podell et al., 2023), GroundedSAM (Ren et al., 2024), InstructPix2Pix (Brooks et al., 2023a)). Furthermore, Image-DPO deliberately corrupts images to produce multiple degraded versions that serve as rejected data in DPO training. Concurrent works (Wang et al., 2024a; Xie et al., 2024) explore similar methods but lack a benchmark to verify enhanced visual focus, fail to establish theoretical connections between their proposed objective and DPO, and utilize limited image transformations (e.g., only randomly cropping). In contrast, we introduce ViLP (Section 3) to assess visual reasoning and provide theoretical foundations (Appendix A), alongside multi-category image corruptions (semantic editing, Gaussian blurring, pixelation).

## 3. ViLP Benchmark

### 3.1. Design Principles

*"What's the tall animal with the longest neck shown?"* Humans readily guess *"giraffe"* based on learned priors, , yet as shown in the bottom-right of Figure 1, it could be an elephant or camel – where visual reasoning corrects the answer. This highlights a potential shortfall in Vision-Language Models (VLMs), which may over-rely on learned visual language priors instead of true visual reasoning, particularly since VLMs are typically fine-tuned on limited image-text data, which is several orders of magnitude smaller than the trained text corpus (Liu et al., 2024a; Tong et al., 2024). Specifically for the scope of visual language priors in this paper, we target **(1)** strong language priors that lead VLMs to derive answers solely from text, and **(2)** potential visual priors causing models to overlook critical uncommon visual cues (e.g., unusual zebra spots in Figure 1).

To evaluate how VLMs handle learned visual language priors, we introduce ViLP, a specialized benchmark of out-of-distribution Question-Image-Answer (QIA) triplets guided by two core principles. First, text-only inference ensures that each question can be answered with high confidence using textual clues alone. Second, visual inference requires that the correct answer—sometimes contradicting common sense—only emerges once an out-of-distribution image is considered. By forcing models to integrate both textual and visual information, ViLP reveals whether they truly engage in visual reasoning or merely rely on memorized patterns.

Mathematically, let $Q$ be a question, $I$ an image, and $A = \{a_{\text{prior}}, \ldots, a_{\text{test}}, \ldots\}$ the set of possible answers. We define $P(a \mid Q)$ as the probability of answer $a$ given $Q$ alone and $P(a \mid Q, I)$ as the probability given both $Q$ and $I$. We consider a prior model $p$, which may represent either human cognition ($P_{\text{human}}$) or a VLM/LLM's learned visual-language prior ($P_\theta$). For constructing our benchmark, we used the following guidances:

**Criterion One**: The question $Q$ alone should strongly favor $a_{\text{prior}}$, where $\delta_1$ is a high-confidence threshold. $a_{\text{prior}}$ usually satisfies common knowledge, such as "soccer ball is a sphere" and "Einstein created $E = mc^2$" (Figure 1).

$$P(a_{\text{prior}} \mid Q) \geq \delta_1 \qquad (1)$$

**Criterion Two**: With the image $I$, the correct answer shifts to $a_{\text{test}}$, where $\delta_2$ is another high-confidence threshold. The image also significantly impacts the answer distribution, where we let $\mathbf{p} = P(\cdot \mid Q)$ and $\mathbf{q} = P(\cdot \mid Q, I)$ denote the answer distributions before and after seeing the image, respectively. Here, $D$ is a divergence measure, and $\delta_3$ is a threshold indicating significant difference. For instance, the image in the 1st row and the 3rd column of Figure 1 turns the answer to *torus*.

$$P(a_{\text{test}} \mid Q, I) \geq \delta_2, \ D_{\text{JS}}(\mathbf{p}, \mathbf{q}) \geq \delta_3 \qquad (2)$$

**Criterion Three**: The answer $a_{\text{test}}$ should be rare and unlikely from $Q$ alone, while $a_{\text{prior}}$ becomes clearly incorrect when considering $I$. This is enforced by a low-confidence threshold $\delta_4$ (e.g., Newton as $a_{\text{test}}$ inferred from the image, contradicting Einstein, shown by the image from the 3rd row and 2nd column of Figure 1).

$$P(a_{\text{test}} \mid Q) \leq \delta_4, \ P(a_{\text{prior}} \mid Q, I) \leq \delta_4 \qquad (3)$$

In designing ViLP, we leverage the human cognition prior $P_{\text{human}}$ as our guiding principle, ensuring each QIA configuration aligns with typical human expectations while requiring visual evidence to override strong textual assumptions. We then compare the learned priors of VLMs and LLMs, denoted $P_\theta$, $P_{\text{human}}$ to evaluate whether these models genuinely engage in visual reasoning rather than relying on memorized patterns.

### 3.2. Question-Image-Answer Generation

Following **Criterion Three**, $a_{\text{test}}$ should be highly improbable based on $Q$ alone yet the correct choice when paired with $I$. Since such images do not exist naturally, we use generative models like DALL·E-3 (Ramesh et al., 2021) and Flux to blend unusual elements that override typical language priors. We incorporate substantial human input and leverage advanced LLMs such as OpenAI-o1 and Claude-3.5-Sonnet to ensure alignment with all the criteria. More details, including text prompts and average cost, are provided in Appendix B.2. Note that as more advanced image generative models—such as the recently introduced 4o image generation—become available, we anticipate generating increasingly abundant and high-quality data for our benchmark, yet future updates will remain consistent with the dataset construction criteria outlined in Section 3.1.

For each question, we design three answers: one $a_{\text{prior}}$ inferred solely from $Q$, and two $a_{\text{test}}$ that defy language priors, requiring visual cues for correctness. We rely on GPT-4 to generate text prompts, produce large-scale images, and then conduct human filtering and refinement. This process faces two main challenges: (1) producing diverse out-of-distribution QA pairs, and (2) synthesizing images that defy specific priors, sometimes necessitating hundreds of samples to find one that accurately matches $Q$ and $a_{\text{test}}$.

Ultimately, we curated 300 questions, each paired with three distinct image-answer sets, totaling 900 QIA triplets. These cover a broad range from low-level recognition (*texture*, *shape*) to high-level

Table 1: **Category Statics**.

| Type | Frequency |
|---|---|
| Texture | 16 |
| Shape | 20 |
| Conceptual combinations | 276 |
| Hallucinated Components | 151 |
| Proverbs | 17 |

reasoning (*conceptual combinations*, *hallucinated components*, *proverbs*). Table 1 summarizes their categorical distribution, with each question spanning an average of 1.6 categories. To reinforce text priors, we present a distractor fact before the question. Rigorous human review ensures that all final QIA triplets are clear and interpretable, as reflected by our human evaluation results in Table 2. Besides, in Appendix B.3, we investigate the impact of image styles by generating more realistic images via 4o image generation and comparing them to those produced by DALL·E-3 (Ramesh et al., 2021) and Flux. We find that realistic images can increase the difficulty of the task, highlighting their importance for future studies.

### 3.3. Dataset Evaluation

All of our questions are designed to elicit single-word answers, an approach that is more efficient and more reliable than sentence-based evaluations that rely on LLM judgment.

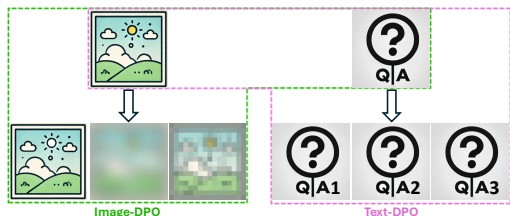

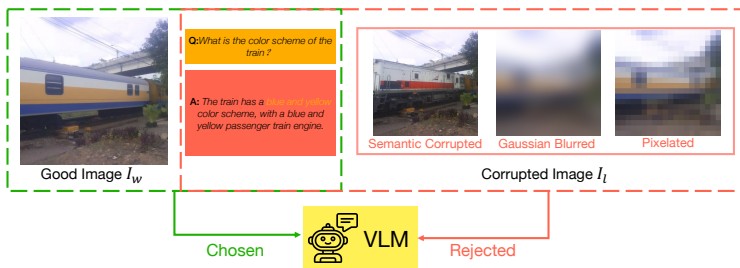

(a) **Image-DPO vs. Text-DPO**: In the green dashed box, we illustrate Image-DPO, which uses a single Q&A pair paired with multiple corrupted images. In contrast, the purple dashed box presents Text-DPO, involving a single input image paired with multiple, distinct Q&A pairs.

(b) **Illustration of Image DPO.** We construct *chosen* and *rejected* pairs by corrupting the image with a set of perturbations while keeping the Q&A unchanged. Perturbations include *semantic editing*, *Gaussian blurring*, and *pixelation*. The mathematical formulations and implementation details are provided in Appendix A and Appendix C, respectively.

By avoiding using LLM, we reduce API fees, computational overhead, and the risk of occasional inaccuracies due to incorrect model reasoning. We explicitly instruct the model to provide a single-word answer, and we evaluate the correctness of each response using a binary system. To ensure a fair evaluation, we devote significant efforts to building a comprehensive set of synonyms and plural for each answer to detect other valid alternative answers. This ensures that the model is only penalized for actual errors, not for providing synonymous or alternative correct responses.

## 4. Image DPO

Inspired by our benchmark, we propose *Image DPO*, a self-improvement method for enhancing VLMs' visual reasoning, featuring a new objective and a data generation pipeline using VLMs themselves and pre-trained image models.

### 4.1. Objective

Existing approaches for VLM self-improvement follow the way used in DPO paper (Rafailov et al., 2024), where the model is trained to distinguish between good and bad answers for a fixed image and question (Figure 2a right). However, this straightforward adaptation may not the best for vision models, as the model sometimes distinguish good and bad answers from the text alone without needing to analyze the image. In contrast, we propose *Image DPO*, a vision-focused objective that creates good and bad question-image-answer pairs by corrupting the image while keeping the question and answer unchanged (Figure 2a left). An example of our synthetic data is illustrated in Figure 2b.

Formally, given an image $I_w$, a question $Q$, and its corresponding answer $A$, we generate a corrupted image $I_l$ via image-editing operations, including Gaussian blur, pixelation, or semantic modifications. The triplet $(Q, I_l, A)$ forms a degraded question-image-answer pair compared to $(Q, I_w, A)$. We train the model to distinguish between good and bad triplets using the objective 4, where $\pi_\theta$ is the target

VLM, $\pi_{\text{ref}}$ is the reference VLM (typically an earlier version of $\pi_\theta$), $S$ is the dataset of good and bad triplets, $\sigma$ is the sigmoid function, and $\alpha$ is a scaling factor. In Proposition 1, we show that objective 4 optimizes an upper bound on the RL fine-tuning objective (Eq. (3) in (Rafailov et al., 2024)).

$$L(\pi_\theta, \pi_{\text{ref}}) = -\mathbb{E}_{Q, I_w, I_l, A \sim S}\Bigg[$$
$$\log \sigma\Big(\alpha \frac{\pi_\theta(A \mid Q, I_w)}{\pi_{\text{ref}}(A \mid Q, I_w)} - \alpha \frac{\pi_\theta(A \mid Q, I_l)}{\pi_{\text{ref}}(A \mid Q, I_l)}\Big)\Bigg]$$
$$(4)$$

Intuitively, since the textual inputs and outputs are identical in both good and bad cases, the gradients of this objective push the model to rely more on the vision branch, driving a shift in gradient direction when processing normal images $I_w$ compared to corrupted images $I_l$ (Figure 6). This behavior encourages the model to focus more on image inputs rather than relying solely on text-based reasoning, thereby enhancing its performance on visual-related tasks. Our experiments demonstrate Image-DPO objective (Eq. 4) outperforms varies self-improve VLM baselines on ViLP.

**Proposition 1.** *Let* $\mathcal{L}_{\text{RLFT}}(\pi_\theta, \pi_{\text{ref}}; \mathcal{S})$ *be the KL-constrained reward maximization objective used in the RL fine-tuning phase (Appendix Eq. 7), where the dataset* $\mathcal{S} = \{(Q, A, I_w, I_l)\}$ *contains good images* $I_w$ *and corrupted images* $I_l$. *Let* $\mathcal{L}_{\text{ImageDPO}}(\pi_\theta, \pi_{\text{ref}}; \mathcal{S})$ *be the objective from Eq. 4, which compares* $(Q, I_w, A)$ *against* $(Q, I_l, A)$. *Then for any policy* $\pi_\theta$ *and reference model* $\pi_{\text{ref}}$, *we have*

$$\mathcal{L}_{\text{RLFT}}(\pi_\theta, \pi_{\text{ref}}; \mathcal{S}) \leq \mathcal{L}_{\text{ImageDPO}}(\pi_\theta, \pi_{\text{ref}}; \mathcal{S}).$$

*Proof Sketch.* Following (Rafailov et al., 2024), we express the optimal KL-constrained policy in terms of a latent reward function. Applying a Bradley–Terry preference model to question-image-answer triplets $(Q, I_w/I_l, A)$ and using Jensen's inequality yields an upper bound whose minimization is equivalent to $\mathcal{L}_{\text{ImageDPO}}(\pi_\theta, \pi_{\text{ref}}; \mathcal{S})$. A full derivation appears in Appendix A.

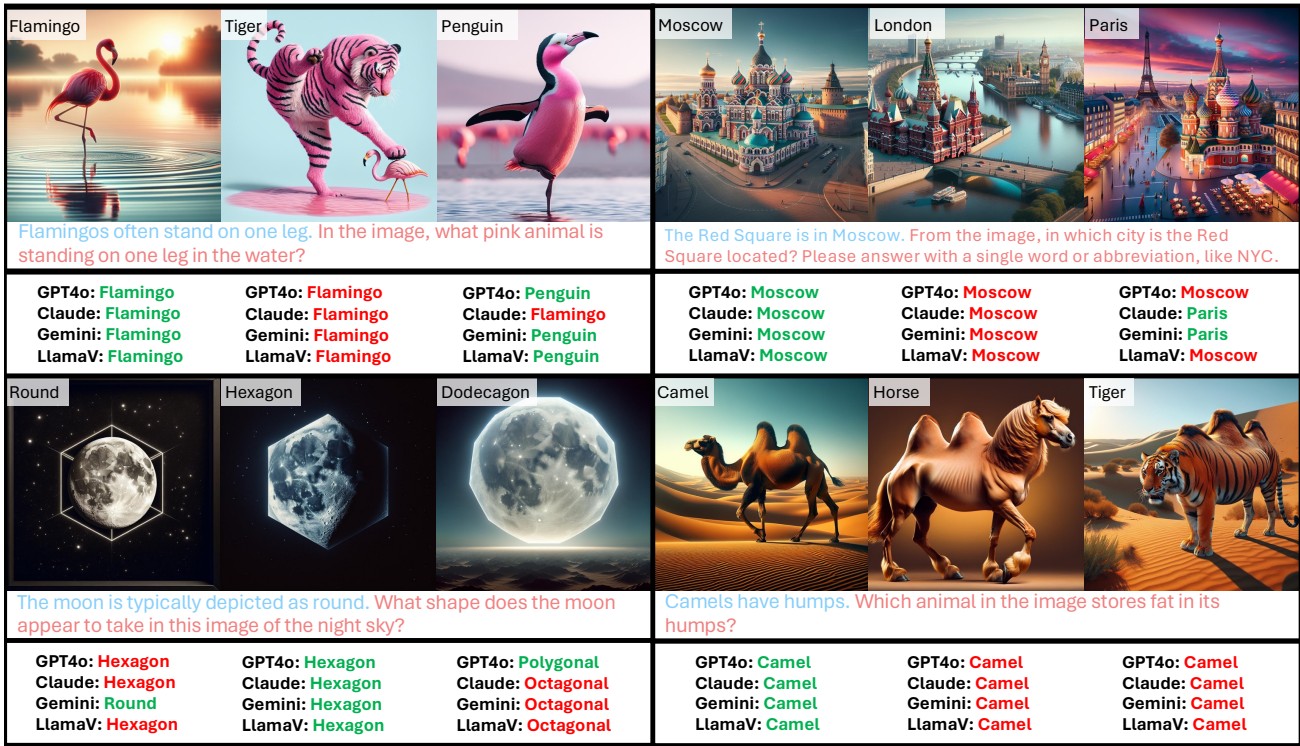

Figure 3: **Qualitative examples.** We show the results from GPT-4o, Claude-3.5-Sonnet, Gemini-1.5-Pro, and Llama-3.2-Vision-90B for some challenging cases. Please refer to Appendix B.6 for categorized failure case analysis.

## 4.2. Data Generation

Training VLMs demands large-scale question-image-answer (QIA) triplets, which are often scarce. To address this, we introduce a scalable data generation pipeline (Appendix Figure 16) that repurposes existing image datasets via VLM themselves and image generative models. Given a seed image from COCO (Lin et al., 2014), Text2VQA (Singh et al., 2019b), or Visual Genome (Krishna et al., 2017), VLMs are tasked with simultaneously selecting appropriate functions (e.g., image generation or editing models) and generating corresponding instructions. These instructions are then used to produce new images, in addition to the seed image, as illustrated in Figure 17. The same VLMs are then employed to generate QA pairs for these newly created images. Following, we apply the mentioned three types of image corruptions to the generated images, constructing good bad pars $(Q, I_w, A)$ and $(Q, I_l, A)$. Specifically, we employ Stable Diffusion XL (Podell et al., 2023; Rombach et al., 2022) for image generation, and use Instruct-Pix2Pix (Brooks et al., 2023a), and Grounded-SAM (Rombach et al., 2022; Ren et al., 2024) for image editing. Example generated data, prompts, and more details are included in Appendix C.

## 5. Experiments

We introduce ViLP, a new benchmark comprising 300 questions. Each question is paired with three unique images

and their corresponding answers—one $QIA_{prior}$ and two $QIA_{test}$—for a total of 900 QIAs. The $QIA_{prior}$ examples (300 in total) align with common language priors (i.e., they can usually be answered correctly by relying on textual cues alone). In contrast, the $QIA_{test}$ examples (600 in total) challenge these priors by requiring visual reasoning.

ViLP features two evaluation settings:

- ViLP$^F$, where both distractor facts and the questions are provided;

- ViLP$^P$, where only the questions themselves are given (i.e., no distractor facts).

We report two metrics in Table 2: average accuracy on $QIA_{test}$ (noted as **Score**) and average accuracy on $QIA_{prior}$ (noted as **Prior**). Our benchmark emphasizes the performance in **Score**.

**Is the QIA easy for humans?** We begin by evaluating our benchmark through a human study. Participants achieved nearly perfect accuracy on ViLP$^F$-Prior and over 98% on ViLP$^F$-Score and ViLP$^P$-Score, confirming that our question-image-answer combinations are unambiguous for human interpretation. Notably, despite $QIA_{test}$ being designed as out-of-distribution examples, humans were still able to correctly distinguish them.

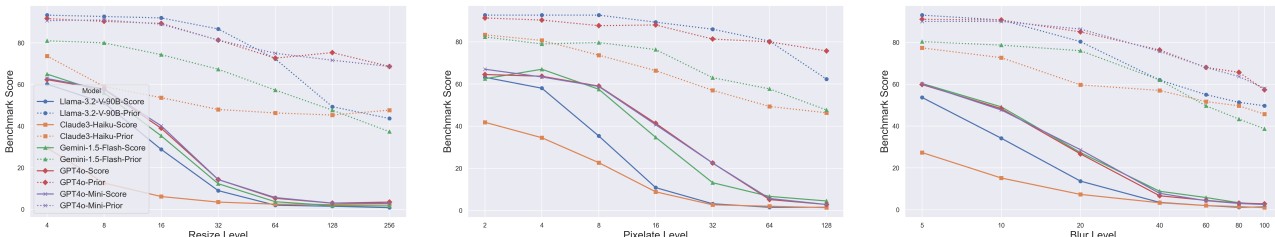

Figure 4: **Comparison of benchmark scores under different image transformations.** Solid line and dotted line refer to ViLP$^F$-Score and ViLP$^F$-Prior, respectively.

Table 2: **Benchmarking on ViLP**. Please refer to the left text for symbol definitions. † indicates the model often fails to follow the instructions.

| Model | ViLP$^F$ | | ViLP$^P$ | |
|---|---|---|---|---|
| | Score | Prior | Score | Prior |
| ***Baseline*** | | | | |
| Human | 98.33 | 99.67 | 98.67 | 96.67 |
| GPT-4o (text only) | 0.0 | 92.33 | 0.17 | 71.33 |
| ***API call only*** | | | | |
| GPT-4o | 66.17 | **91.00** | 56.00 | **87.67** |
| GPT-4V | 57.67 | 88.33 | 38.33 | 85.33 |
| GPT-4o-Mini | 57.67 | 89.00 | 46.67 | 84.67 |
| Claude-3.5-Sonnet | **70.00** | 84.33 | 59.33 | 86.67 |
| Claude-3-Opus | 59.17 | 74.00 | 43.00 | 82.67 |
| Claude-3-Sonnet | 48.83 | 83.67 | 40.33 | 81.33 |
| Claude-3-Haiku | 43.67 | 82.67 | 34.83 | 82.33 |
| Gemini-1.5-Pro | 60.50 | 79.33 | 48.00 | 83.00 |
| Gemini-1.5-Flash | 54.50 | 83.33 | **69.17** | 79.67 |
| ***Open weights*** | | | | |
| Llama-3.2-Vision-11B | **67.33** | 76.67 | 61.17 | 79.33 |
| Llama-3.2-Vision-90B | 64.00 | 91.67 | **63.17** | **83.33** |
| MolmoE-1B | 48.67 | 57.33 | 47.83 | 69.00 |
| Molmo-7B-O | 57.83 | 60.67 | 47.33 | 76.33 |
| Molmo-7B-D | 54.5 | 69.00 | 46.17 | 72.33 |
| Molmo-72B | 60.33 | 85.00 | 47.17 | 82.33 |
| Qwen2-VL-7B | 50.50 | 83.00 | 48.67 | 80.33 |
| Qwen2-VL-72B | 56.50 | **92.33** | 53.83 | 83.00 |
| InternVL2-8B | 47.00 | 66.67 | 43.00 | 75.00 |
| InternVL2-76B | 42.67 | 47.67 | 50.84 | 74.33 |
| LLaVA-1.5-7B | 29.67 | 71.33 | 37.67 | 65.67 |
| LLaVA-1.5-13B | 35.33 | 81.00 | 41.50 | 73.67 |
| Cambrian-1-8B† | 8.67 | 43.67 | 32.50 | 63.67 |
| LLaVA-OneVision-7B | 54.17 | 82.33 | 49.67 | 75.00 |
| LLaVA-OneVision-72B † | 1.67 | 3.00 | 5.22 | 11.67 |

Humans performed slightly better on ViLP$^F$-Prior when distractor facts were provided, as they could easily identify that these facts aligned with the correct answers. Moreover, ViLP$^F$-Score was marginally lower when facts were introduced, as the distractor facts added some noise and caused minor confusion, although the impact of this noise is relatively small. These findings are consistent with the design principles of our benchmark.

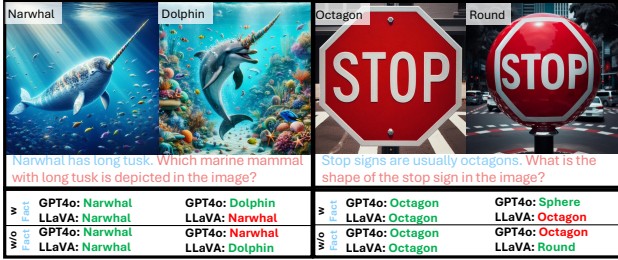

Figure 5: **Qualitative results before and after removing** distractor facts. GPT-4o and LLaVA-1.5-13B models yield completely opposite behaviors.

**Are our QIAs aligned with the learned priors of VLMs?** We tested GPT-4o (text only) on our questions (removing all image references). Despite no visual content, it correctly answered 92.33% on ViLP$^F$-Prior. The accuracy drops to 71.33% once distractor facts are removed, showing that these facts significantly guide the answer. For QIA$_{test}$, GPT-4o (text only) accuracy nearly falls to 0% (ViLP$^F$-Score & ViLP$^P$-Score), indicating the QIA$_{test}$ cannot be answered using text alone.

**How do VLMs perform on our benchmark?** Although our benchmark questions are distinguishable for humans, they are challenging for VLMs. Even the advanced VLM models like GPT-4o, have a clear performance gap (66.17% v.s. 98.33%) compared to humans' performance on ViLP$^F$-Score, indicating the difficulty of these questions for VLMs. Claude-3.5-Sonnet achieved the best score 70%, while most of the commercial VLMs are below 60%. Figure 3 highlights sample outputs from top commercial and open-source models, including GPT-4o, Claude-3.5-Sonnet, Gemini-1.5-Pro, and Llama-3.2-Vision-90B. They face significant challenges when addressing these cases in our ViLP, whereas humans can arrive at correct answers after consideration. Notably, it is encouraging to see that some open-source models achieved over 60% accuracy on ViLP$^F$-Score, with performance nearing that of their commercial counterparts, including Llama-3.2-Vison and Molmo-72B. Additionally, we provide more detailed failure-case analysis in Appendix B.6, where we group the errors into six recurring modes—(i) shape-recognition slips, (ii) counting mistakes, (iii) prior-

Table 3: **Effectiveness of Image-DPO on General VQA benchmarks.**

| VLMs | $\text{ViLP}^{\text{F}}_{\text{Score}}$ | $\text{ViLP}^{\text{P}}_{\text{Score}}$ | $\text{NB}_{\text{Q}}$ | $\text{NB}_{\text{I}}$ | $\text{NB}_{\text{G}}$ | $\text{NB}_{\text{B}}$ | MM-Vet | $\text{CHAIR}^{\text{S}}\downarrow$ | $\text{CHAIR}^{\text{I}}\downarrow$ |
|---|---|---|---|---|---|---|---|---|---|
| LLaVA-1.5-7B | 29.67 | 37.67 | 37.7 | 43.8 | 12.7 | 67.3 | 31.1 | 49.1 | 14.8 |
| LLaVA-1.5-7B + Image-DPO | $34.17^{\uparrow 4.5}$ | $39.33^{\uparrow 1.66}$ | $39.79^{\uparrow 2.09}$ | $45.47^{\uparrow 1.67}$ | $14.16^{\uparrow 1.46}$ | $68.45^{\uparrow 1.15}$ | $32.3^{\uparrow 1.2}$ | $45^{\uparrow -4.1}$ | $12.3^{\uparrow -2.5}$ |
| LLaVA-1.5-13B | 35.33 | 41.5 | 39.6 | 44.6 | 14.8 | 68.9 | 36.1 | 48.3 | 14.1 |
| LLaVA-1.5-13B + Image-DPO | $38.17^{\uparrow 2.84}$ | $42.5^{\uparrow 1}$ | $42.68^{\uparrow 3.08}$ | $47.37^{\uparrow 2.77}$ | $17.16^{\uparrow 2.36}$ | $70.36^{\uparrow 1.46}$ | $37.5^{\uparrow 1.4}$ | $42.6^{\uparrow -5.7}$ | $11.6^{\uparrow -2.5}$ |
| Cambrian-8B | 8.67 | 32.5 | 44.6 | 47.9 | 19.4 | 71.5 | 51.4 | 14.5 | 4.7 |
| Cambrian-8B + Image-DPO | $20.83^{\uparrow 12.16}$ | $39.3^{\uparrow 6.83}$ | $46.5^{\uparrow 1.9}$ | $50.2^{\uparrow 2.3}$ | $20^{\uparrow 0.6}$ | $72^{\uparrow 0.5}$ | $51.7^{\uparrow 0.3}$ | $11.4^{\uparrow -3.1}$ | $4.4^{\uparrow -0.3}$ |

Table 4: **Benchmark Performance with More Fine-Grained Categories.** We extended our evaluation system to include three categories—*Correct*, *Wrong*, and *Invalid*. "Invalid" responses (not single-word) are further split into *Invalid-Correct* (noted as Invalid-✓) and *Invalid-Wrong* (noted as Invalid-✗) based on GPT assessment.

| Model | $\text{ViLP}^{\text{F}}$ | | | | | | | | $\text{ViLP}^{\text{P}}$ | | | | | | | |
|---|---|---|---|---|---|---|---|---|---|---|---|---|---|---|---|---|
| | Score | | | | Prior | | | | Score | | | | Prior | | | |
| | Correct | Wrong | Invalid-✓ | Invalid-✗ | Correct | Wrong | Invalid-✓ | Invalid-✗ | Correct | Wrong | Invalid-✓ | Invalid-✗ | Correct | Wrong | Invalid-✓ | Invalid-✗ |
| Cambrian-8B | 8.67% | 28.83% | 29.17% | 33.33% | 43.67% | 2.33% | 54.00% | 6.00% | 32.50% | 37.50% | 11.67% | 18.33% | 63.67% | 8.33% | 22.00% | 6.00% |
| LLaVA-OneVision-7B | 54.17% | 42.83% | 1.33% | 1.67% | 82.33% | 15.33% | 2.33% | 0.00% | 49.67% | 47.00% | 1.33% | 2.00% | 75.00% | 20.33% | 3.67% | 1.00% |
| LLaVA-OneVision-72B | 1.67% | 1.00% | 43.33% | 54.00% | 3.00% | 0.33% | 90.33% | 6.33% | 5.22% | 5.78% | 39.00% | 50.00% | 11.67% | 1.33% | 78.67% | 8.33% |
| GPT-4o | 66.17% | 33.77% | 0.03% | 0.03% | 91.00% | 8.33% | 0.33% | 0.33% | 56.00% | 43.17% | 0.50% | 0.03% | 87.67% | 11.33% | 1.00% | 0.00% |

biased rejection of counterfactual scenes, (iv) proverb-driven hallucinations, (v) misclassification of creative concepts, and (vi) text-over-vision errors on blended images.

**Do distractor facts really distract?** In ViLP$^{\text{F}}$ setting, we add a distractor fact before the question. Since these facts implicitly suggest incorrect answers for QIA$_{\text{test}}$, we expected this change to make the questions more suggestive and lower the ViLP$^{\text{F}}$-Score, as the distractors would mislead the VLMs. Surprisingly, GPT-4o benefits from distractor facts, improving accuracy on QIA$_{\text{test}}$. We hypothesize these facts highlight question focus, narrowing the search space. However, weaker models like LLaVA-1.5-13B (Liu et al., 2023b) often get misled by the distractors, hurting their *Score* but boosting *Prior*. For instance, as shown in Figure 5, with including distractor facts, LLaVA-1.5-13B consistently predicts the distractor fact as the answer. However, once the distractors are removed, it can then predict correctly.

For bad instruction-following models like Cambrian-8B (Tong et al., 2024), distractor facts significantly hinder adherence to explicit instructions, such as providing single-word answers. With facts, Cambrian-8B fails to follow instructions in 62% of cases, compared to 30% without (a nearly 2x increase). Manual review shows 59% of these failures are contextually correct, yielding an adjusted accuracy of 47.92%. Similarly, LLaVA-OneVision-72B (Li et al., 2024b) often generates detailed analyses despite explicit single-word prompts. This trend highlights a concerning trend: focusing on improving performance on well-established benchmarks may come at the cost of basic instruction-following abilities, ultimately limiting the practical utility of these models in real-world applications.

**How image transformations affect the results?** We also investigate how image transformations, including resizing, Gaussian blur, and pixelation, affect ViLP performance. The results, shown in Figure 4, reveal that the ViLP$^{\text{F}}$-Score rapidly decreases as the severity of the transformations (x-axis) increases, while the ViLP$^{\text{F}}$-Prior score remains around 50%. Interestingly, GPT-4o, when using degraded images, performs worse in ViLP$^{\text{F}}$-Prior than when no images are used, i.e., GPT-4o (text only) in Table 2.

**Comparison of ViLP with other VQA datasets:** To highlight the distinctions between ViLP and existing benchmarks, including Winoground (Thrush et al., 2022), Whoops! (Bitton-Guetta et al., 2023a), and Hallucination-Bench (Guan et al., 2023), we conduct a comparative analysis of both their high-level design principles and low-level data formats. This comparison incorporates qualitative and quantitative insights as detailed in Appendix B.4.

**Finer-Grained Evaluation** Currently, we explicitly ask VLMs to produce a single-word response for evaluation, and penalize responses that include long sentences, as we believe instruction-following is a foundation capability of VLMs—thus, failure to comply warrants a penalty. Although this evaluation protocol is *cost-efficient without calling LLM*, *fast*, and *reliable*, it may conflate semantic errors with instruction-following failures.

To address this, we introduce an optional finer-grained evaluation protocol with three categories: *Correct*, *Wrong*, and *Invalid*. *Correct* and *Wrong* denote semantically accurate or inaccurate answers that comply with the single-word constraint. *Invalid* captures responses that violate the format. For these, we further offer an optional GPT-based classification into *Invalid-Correct* or *Invalid-Wrong*, providing greater granularity at the cost of increased evaluation overhead. We show these evaluation results in Table 4.

Commercial models such as GPT-4o follow instructions reliably. In contrast, LLaVA-OneVision-72B matches other models in reasoning ability (*Correct + Invalid-Correct*) yet ignores the single-word instruction in most instances. Cambrian-8B similarly yields a significant number of *Invalid* cases. Counterintuitively, LLaVA-OneVision-72B performs markedly worse at instruction following than its smaller counterpart, LLaVA-OneVision-7B, and shows a stronger bias toward learned priors—reflected in its lower ViLP Score and higher ViLP Prior. We hypothesize that instruction following offers an additional lens for probing prior reliance: models dominated by priors tend to generate verbose, prior-driven answers (*i.e.*, long responses) that disregard explicit single-word directives.

### 5.1. Image DPO

In this section, we evaluate *Image-DPO* (Section 4.1) on both ViLP and general VQA benchmarks. As an **ablation baseline**, we introduce *Text-DPO*, which uses the same Question-Image-Answe (QIA) generation process as we used in Image-DPO but applies LLM self-rewarding objective (Yuan et al., 2024) (the standard DPO objective). In Text-DPO, good and bad pairs stem from VLM-generated positive and negative answers, while the question and image remain fixed. As shown in the right of Figure 2a, the green box depicts Image-DPO, generating corrupted images via semantic edits, Gaussian blur, and pixelation while keeping the question and answer constant; the purple box illustrates Text-DPO, which fixes the image and varies the answers with associated ratings. This setup parallels other VLM self-rewarding work (Zhou et al., 2024a; Deng et al., 2024; Zhou et al., 2024b; Wang et al., 2024c;a).

For baselines, we compare with VLM self-improvement methods, including SIMA (Wang et al., 2024c), HADPO (Zhao et al., 2023a), and EOS (Yue et al., 2024), by using their publicly available checkpoints. Additionally, we train models using the dataset and code provided in RLHF-V (Yu et al., 2023a) and V-DPO (Xie et al., 2024). All models use LLaVA-7B for a comprehensive comparisons as many paper only release 7B checkpoints. Table 5 shows that Image-DPO achieves the highest across all the metrics.

Besides, we evaluate the proposed Image-DPO algorithm across three VLM models—Cambrian-8B, LLaVA-1.5-7B, and LLaVA-1.5-13B—using several popular VLM benchmarks that focus on different aspects, including compositionality & biases (NaturalBench (Li et al., 2024a)), general visual reasoning (MM-Vet (Yu et al., 2023c)), and hallucinations (CHAIR (Rohrbach et al., 2018)). The results, presented in Table 3, show consistent performance improvements across both datasets and models, further demonstrating the effectiveness of our Image DPO method.

Table 5: **Comparisons of Image-DPO on ViLP.**

| Model | ViLP F | | ViLP P | |
|---|---|---|---|---|
| | Score | Prior | Score | Prior |
| LLaVA-1.5-7B | 29.67 | 71.33 | 37.67 | 65.67 |
| +HADPO (Zhao et al., 2023b) | 33.00 | 74.33 | 38.50 | 65.00 |
| +RLHF-V (Yu et al., 2023a) | 29.50 | 75.00 | 36.33 | 65.33 |
| +EOS (Yue et al., 2024) | 31.33 | 67.00 | 38.67 | 65.67 |
| +SIMA (Wang et al., 2024b) | 27.83 | 68.67 | 36.17 | 66.00 |
| +V-DPO (Xie et al., 2024) | 29.50 | 72.67 | 37.83 | 67.67 |
| +Text-DPO | 31.34 | 71.67 | 37.83 | 65.67 |
| +Image-DPO | **34.17** | **75.00** | **39.33** | **68.00** |

**TextDPO on Corrupted Images.** Does Image-DPO's improvement stem from the objective itself, or is it merely due to training on more perturbed data? To investigate this, we conduct an ablation where we train TextDPO with corrupted images. Specifically, for each pair of $(Q, I, A_w)$ and $(Q, I, A_l)$ used in Text-DPO, we apply the same corruptions as Image-DPO to $I$ to form $(Q, I', A_w)$ and $(Q, I', A_l)$. Results show that LLaVA-v1.5-7B trained with this TextDPO variant achieves a ViLP$^F$-Score of 31, a ViLP$^P$-Score of 37.5, and 30.3 on MMVET. In contrast, ImageDPO outperforms it across all metrics, achieving a ViLP$^F$-Score of 34.17, a ViLP$^P$-Score of 39.33, and 32.3 on MMVET.

**Impact of Synthetic Image Augmentation.** When we remove the image-generation stage and fine-tune *ImageDPO* only on QIA pairs drawn from the seed datasets (COCO, VG, Text2VQA), performance slips across *every* metric: ViLP$^F$Score declines from 34.17 to 31.17 and ViLP$^P$Score from 39.33 to 38.00; the four novel-bias scores (NB$_Q$, NB$_I$, NB$_G$, NB$_B$) each lose roughly one point; MM-Vet drops from 32.3 to 31.4; and hallucination rates rise (CHAIR$^S$↓: $45 \rightarrow 47$, CHAIR$^I$↓: $12.3 \rightarrow 13.6$). These uniform declines confirm that synthetic images by image generative models broaden the visual distribution, inject the priors of powerful generative models, and are crucial for achieving the full gains of ImageDPO.

## 6. Conclusion

In conclusion, we present the ViLP benchmark to probe the challenge of visual language bias in Vision-Language Models (VLMs). By utilizing advanced image generation models and designing questions that demand visual cues for accurate responses, our benchmark includes images that defy language priors, revealing the limitations of current VLMs. Our method, Image-DPO, which incorporates self-generated VQA pairs and image corruption for training, has demonstrated promising improvements in enhancing visual reliance, as evidenced by performance gains on open-source VLMs, such as LLaVA-v1.5 and Cambrian.

# Acknowledgment

This work was supported by the LG AI Research grant. We also extend our gratitude to the OpenAI Researcher Access Program for providing credits used to access OpenAI's APIs. Additionally, we appreciate Chris Rockwell and Jiaming Yang for their contributions to our human evaluation studies, and thank Yongyi Yang for discussions regarding the proposition proof.

# Impact Statement

This work aims to advance the field of Vision-Language Models (VLMs) by diagnosing and mitigating their reliance on learned visual language priors. By introducing the ViLP benchmark and Image-DPO training framework, our contributions promote more robust and interpretable visual reasoning in AI systems. While ViLP dataset involves synthetic image generation, we have rigorous human review to avoid generating harmful or deceptive content. Our findings may improve AI alignment with human perception, benefiting applications in accessibility, education, and AI-assisted decision-making.

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

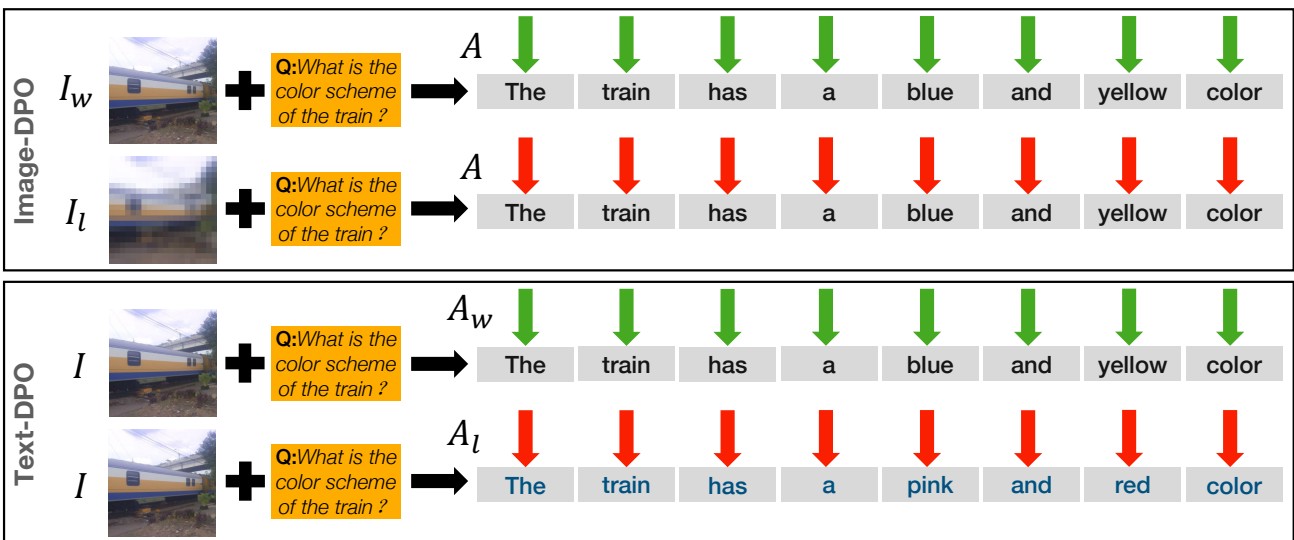

Figure 6: **Gradients difference between Image-DPO and Text-DPO.** For Text-DPO (Rafailov et al., 2024), the model receives positive gradients (green arrows) for the preferred answer $A_w$ and negative gradients $A_l$ (red arrows) for the dispreferred answer. In contrast, our proposed Image-DPO approach applies positive gradients when the preferred image $I_w$ is input and negative gradients for the dispreferred image $I_l$, both based on the same output answer.

## A. Image-DPO Mathematical Details

In this section, we give the complete proof of Image DPO. Simlarly to DPO, we start from the RL fine-tuning objective and then derive its variant for image dpo.

### A.1. RL Fine-tuning for VLM

**SFT** Given question $Q$, answer $A$ and image $I$, we can train a SFT model $\pi^{SFT}$ with supervised learning on high-quality data. As the SFT model is a language generation model, it is still a model modeling the text outputs with question and image. $\pi^{SFT}(A|Q, I)$

**Reward Modeling Phase** In this stage, we construct a static dataset of comparisons $\mathcal{S} = \{A^i, Q^i, I_w^i, I_l^i\}$, and we present the QIA pairs $(Q, I_w, A)$, $(Q, I_l, A)$ to human for preference.

Following the idea of RL fine-tuning, the preference are assumed to be obtained from a a latent reward function $r^*(Q, I, A)$ which are not tractable, and we we use BT model to represent the preference distribution $p^*$ as:

$$p^*((Q, I_w, A) \succ (Q, I_l, A)) = \frac{\exp\left(r^*\left(Q, I_w, A\right)\right)}{\exp(r^*(Q, I_w, A) + \exp(r^*(Q, I_l, A))} \tag{5}$$

Now given the human labeled preference, we can try to optimize a reward model $r_\phi$ to estimate $r^*$ by using maximum likelihood. Framing this as a binary classification, we can have this negative log-likelihood loss:

$$\mathcal{L}_R(r_\phi, \mathcal{S}) = -\mathbb{E}_{(A,Q,I_w,I_l)\sim\mathcal{S}}\left[\log \sigma(r_\phi(Q, I_w, A) - r_\phi(Q, I_l, A))\right] \tag{6}$$

Here $\sigma$ is a logistic function. Basically, this reward function gives score jointly considering image, question and image quality.

**RL Fine-Tuning Phrase**

During the RL phase, the learned reward function is used to provide feeback to the VLM model. Following DPO

paper (Rafailov et al., 2024), the optimization is formulated as :

$$\max_{\pi_\theta} \mathbb{E}_{(Q,I)\sim\mathcal{S},A\sim\pi_\theta(A|Q,I)} \left[r_\phi(Q,I,A)\right] - \beta \mathbb{D}_{\text{KL}} \left[\pi_\theta(A|Q,I)\|\pi_{\text{ref}}(A|Q,I)\right], \tag{7}$$

where $\beta$ is a parameter controlling the deviation from the base reference policy $\pi_{\text{ref}}$, namely the initial SFT model $\pi^{\text{SFT}}$. Due to the discrete nature of language generation, this object is also not differentiable and is typically optimized with reinforcement learning (Jaques et al., 2017; 2020).

### A.2. Image DPO and RL Fine-Tuning

According to the DPO paper, a straightforward optimal solution to the KL-constrained reward function maximization object in Eq. 6 is:

$$\pi_r(A|Q,I) = \frac{1}{Z(Q,I)}\pi_{\text{ref}}(A|Q,I)\exp(\frac{1}{\beta}r(Q,I,A)) \tag{8}$$

where $Z(Q,I) = \sum_A \pi_{\text{ref}}(A|Q,I)\exp(\frac{1}{\beta}r(Q,I,A))$ is a partition function. Here $r$ should be any reward function, which makes $Z$ hard to tract. We provide the proof of this step in A.3.

Taking the logarithm of both side, and with some algebra, we get

$$r(Q,I,A) = \beta\frac{\pi_r(A|Q,I)}{\pi_{\text{ref}}(A|Q,I)} + \beta\log Z(Q,I) \tag{9}$$

This parametrization could be applied to ground-truth reward $r^*$ and the corresponding optimal model $\pi^*$.

The BT model with the optimal policy is

$$p^*((Q,I_1,A) \succ (Q,I_2,A)) = \frac{\exp\left(r^*(Q,I_w,A)\right)}{\exp\left(r^*(Q,I_w,A)\right) + \exp\left(r^*(Q,I_l,A)\right)} \tag{10}$$

We plug Eq. 9 into the BT model, we have:

$$
\begin{aligned}
p^*((Q,I_1,A) \succ (Q,I_2,A)) &= \frac{\exp\left(\beta\log\frac{\pi^*(A|Q,I_w)}{\pi_{\text{ref}}(A|Q,I_w)} + \beta\log Z(Q,I_w)\right)}{\left(\beta\log\frac{\pi^*(A|Q,I_w)}{\pi_{\text{ref}}(A|Q,I_w)} + \beta\log Z(Q,I_w)\right) + \left(\beta\log\frac{\pi^*(A|Q,I_l)}{\pi_{\text{ref}}(A|Q,I_l)} + \beta\log Z(Q,I_l)\right)} \\
&= \frac{1}{1 + \exp(\beta\log\frac{\pi^*(A|I_l,Q)}{\pi_{\text{ref}}(A|I_l,Q)} - \beta\log\frac{\pi^*(A|I_w,Q)}{\pi_{\text{ref}}(A|I_w,Q)} + \beta\log Z(I_l,Q) - \beta\log Z(I_w,Q))} \\
&= \sigma\left(\exp(\beta\log\frac{\pi^*(A|I_l,Q)}{\pi_{\text{ref}}(A|I_l,Q)} - \beta\log\frac{\pi^*(A|I_w,Q)}{\pi_{\text{ref}}(A|I_w,Q)} + \beta\log Z(I_l,Q) - \beta\log Z(I_w,Q))\right)
\end{aligned}
$$

Now we have the probability of human preference data in terms of the optimal policy rather than the reward model, we can formulate a maximum likelihood objective for a policy $\pi_\theta$. Our policy objective is :

$$\mathcal{L}(\pi_\theta;\pi_{\text{ref}}) = \mathbb{E}_{(Q,A,I_w,I_l)\sim\mathcal{S}}\left[-\log\sigma\left(\beta\log\frac{\pi_\theta(A|I_w,Q)}{\pi_{\text{ref}}(A|I_w,Q)} - \beta\log\frac{\pi_\theta(A|I_l,Q)}{\pi_{\text{ref}}(A|I_l,Q)} + \beta\log Z(I_w,Q) - \beta\log Z(I_l,Q)\right)\right] \tag{11}$$

As $f(x) = -\log\sigma(x)$ is a convex function ($\sigma$ is the sigmoid function), we can apply Jensen's inequality $f(\frac{1}{2}x + \frac{1}{2}y) \leq \frac{1}{2}f(x) + \frac{1}{2}f(y)$:

$$\mathcal{L}(\pi_\theta;\pi_{\text{ref}}) \leq \mathbb{E}\left[-\frac{1}{2}\log\sigma\left(2\beta\log\frac{\pi_\theta(A|I_w,Q)}{\pi_{\text{ref}}(A|I_w,Q)} - 2\beta\log\frac{\pi_\theta(A|I_l,Q)}{\pi_{\text{ref}}(A|I_l,Q)}\right) - \frac{1}{2}\log\sigma\left(2\beta\log Z(I_w,Q) - 2\beta\log Z(I_l,Q)\right)\right] \tag{12}$$

As $\log \sigma (Z(I, Q))$ is not a function of $\pi_\theta$, the above objective is equivalent to the below Eq.13, where $\alpha = 2\beta$. It is the same as our objective listed in Eq.4 of the main paper.

$$\mathcal{L}(\pi_\theta; \pi_{\text{ref}}) \leq -\mathbb{E}_{(Q, I_w, I_l, A) \sim S} \left[ \log \sigma(\alpha \frac{\pi_\theta(A \mid Q, I_w)}{\pi_{\text{ref}}(A \mid Q, I_w)} - \alpha \frac{\pi_\theta(A \mid Q, I_l)}{\pi_{\text{ref}}(A \mid Q, I_l)}) \right] \tag{13}$$

In this sense, our optimization objective Eq.4 in main paper are optimizing the upper bound on the RL fine-tuning objective, i.e., Eq.7.

### A.3. Deriving the Optimum of the KL-Constrained Reward Maximization Objective

In this appendix, we will derive Eq.8. Similarly to Eq.7, we optimize the following objective:

$$\max_\pi \mathbb{E}_{(Q, I) \sim \mathcal{S}, A \sim \pi} [r(Q, I, A)] - \beta D_{\text{KL}} [\pi(A|Q, I) \| \pi_{\text{ref}}(A|Q, I)] \tag{14}$$

under any reward function $r(Q, I, A)$, reference model $\pi_{\text{ref}}$, and a general non-parametric policy class. We now have:

$$
\begin{aligned}
&\max_\pi \mathbb{E}_{(Q, I) \sim \mathcal{S}, A \sim \pi} [r(Q, I, A)] - \beta D_{\text{KL}} [\pi(A|Q, I) \| \pi_{\text{ref}}(A|Q, I)] \\
&= \max_\pi \mathbb{E}_{(Q, I) \sim \mathcal{S}} \mathbb{E}_{A \sim \pi(A|Q, I)} \left[ r(Q, I, A) - \beta \log \frac{\pi(A|Q, I)}{\pi_{\text{ref}}(A|Q, I)} \right] \\
&= \min_\pi \mathbb{E}_{(Q, I) \sim \mathcal{S}} \mathbb{E}_{A \sim \pi(A|Q, I)} \left[ \log \frac{\pi(A|Q, I)}{\pi_{\text{ref}}(A|Q, I)} - \frac{1}{\beta} r(Q, I, A) \right] \\
&= \min_\pi \mathbb{E}_{(Q, I) \sim \mathcal{S}} \mathbb{E}_{A \sim \pi(A|Q, I)} \left[ \log \frac{\pi(A|Q, I)}{\frac{1}{Z(Q, I)} \pi_{\text{ref}}(A|Q, I) \exp \left( \frac{1}{\beta} r(Q, I, A) \right)} - \log Z(Q, I) \right]
\end{aligned}
\tag{15}
$$

where we have the partition function:

$$Z(Q, I) = \sum_A \pi_{\text{ref}}(A|Q, I) \exp \left( \frac{1}{\beta} r(Q, I, A) \right) \tag{16}$$

Observe that the partition function depends solely on $(Q, I)$ and the reference policy $\pi_{\text{ref}}$, and is independent of the policy $\pi$. We can now define the Equation 8.

$$\pi^*(A|Q, I) = \frac{1}{Z(Q, I)} \pi_{\text{ref}}(A|Q, I) \exp \left( \frac{1}{\beta} r(Q, I, A) \right), \tag{17}$$

# B. More details and comparisons of our benchmarks

## B.1. More data samples of ViLP

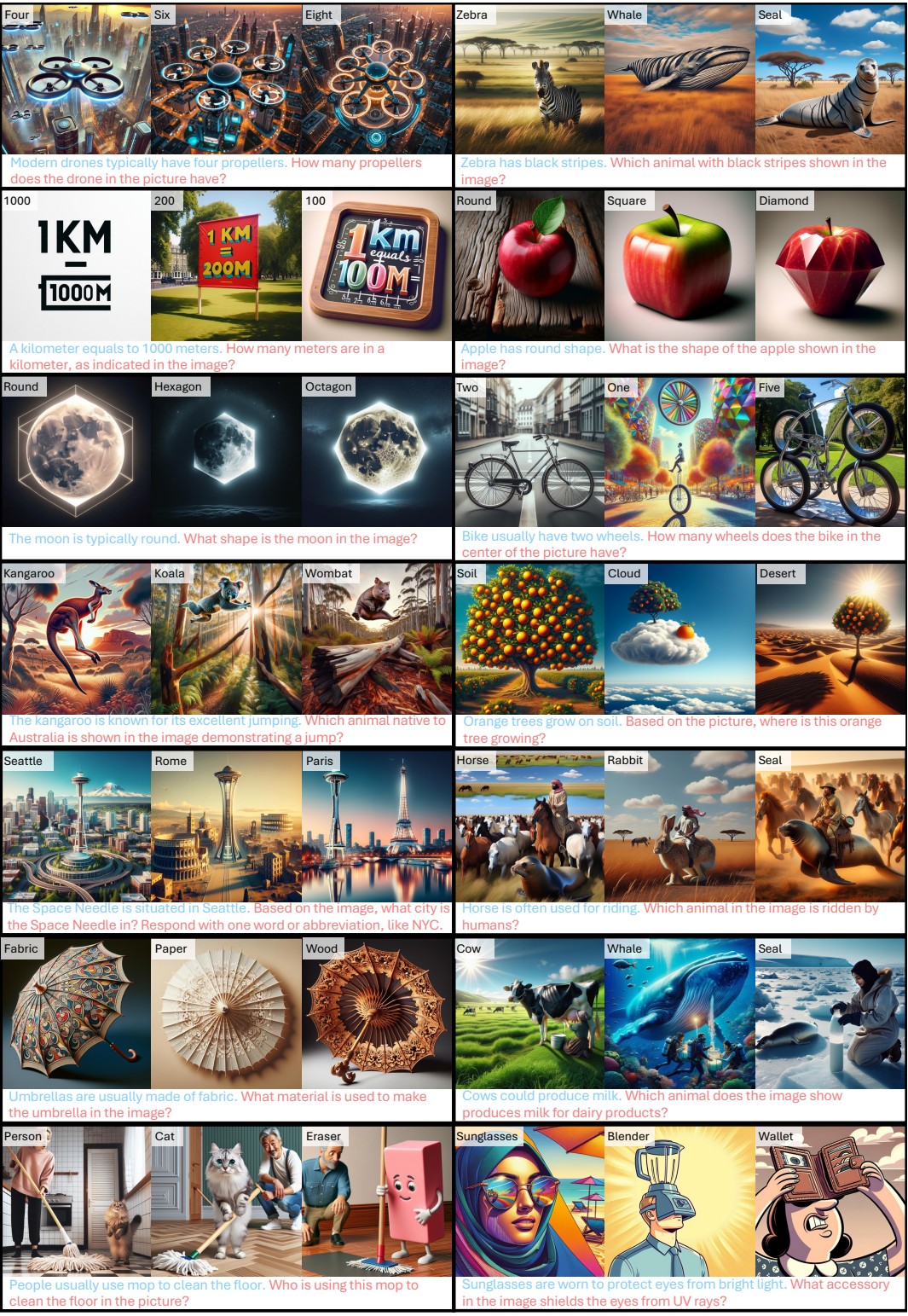

Figure 7: **Randomly sampled data from ViLP.**

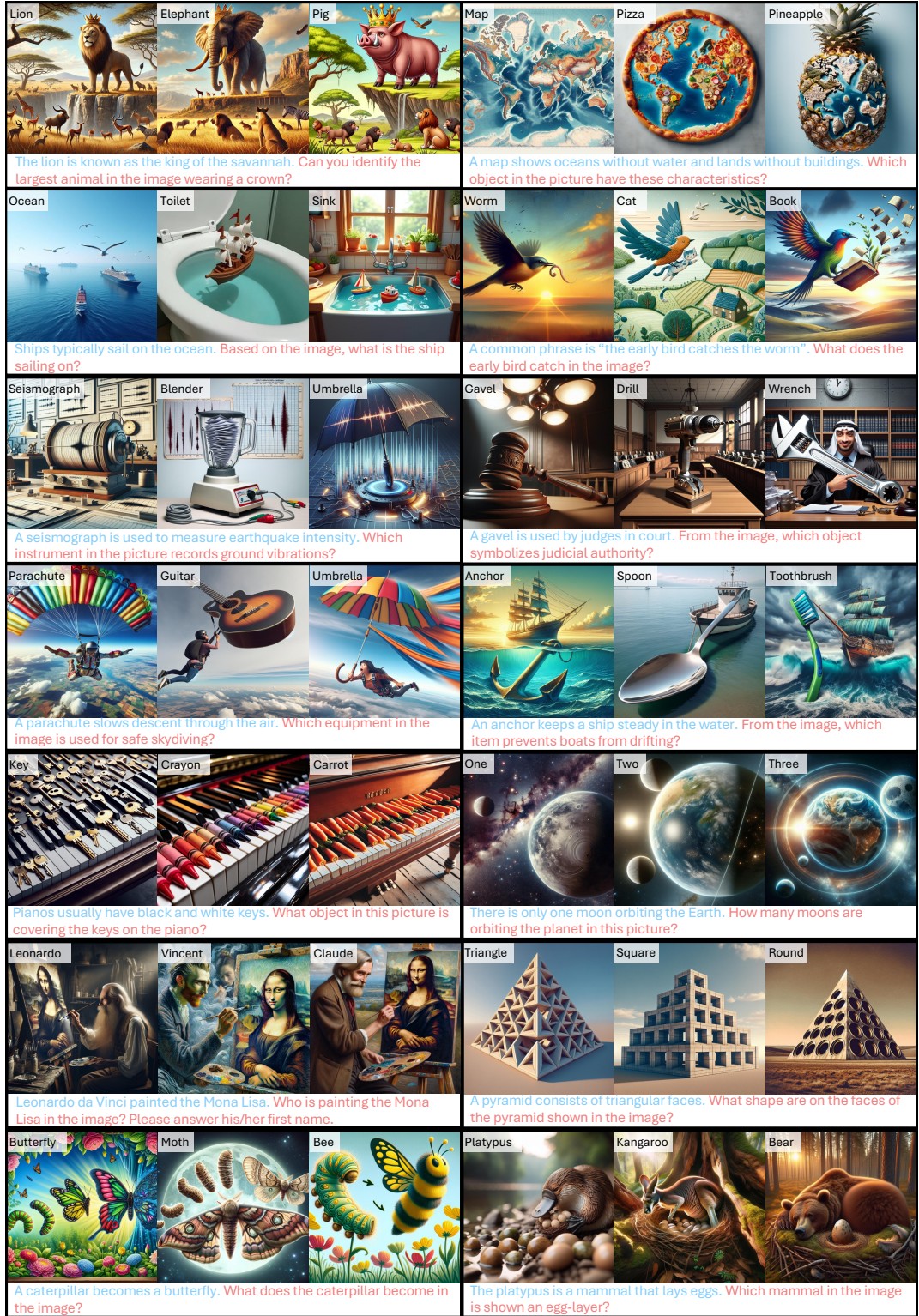

Figure 8: **Randomly sampled data from ViLP.**

## B.2. More details in ViLPbenchmark data generation

Our proposed dataset introduces Question-Image-Answer (QIA) triplets designed to challenge state-of-the-art Vision-Language Models (VLMs) against visual language priors. The construction process combines human-guided and automated efforts to ensure quality and alignment.

**Question-Answer (QA) Generation:** Most QA pairs are authored by humans following the design principles in Section 3.1. Additionally, candidate QA pairs are generated using models like OpenAI-O1 and Claude-3.5-Sonnet with carefully crafted prompts. One example prompts shown in Figure 10. These candidates undergo human review, where they are refined or removed to meet our quality standards.

**Image Generation:** For each QA pair, we use GPT-4 to generate multiple descriptive image prompts (see Figure 9). These prompts are provided to image generation models, such as FLUX and DALL-E 3, to produce candidate images. Human reviewers then select the most suitable image or request re-generation as needed to maintain consistency with the QA context.

**Human Review and Testing:** At every stage, human reviewers rigorously evaluate the generated outputs to ensure quality, clarity, and challenge level. In addition to filtering out low-quality or insufficiently challenging triplets, we dynamically test the QIAs to confirm that they remain intuitive for humans while being difficult for VLMs.

**Cost:** The complexity of our data creation process leads to a significant average cost of approximately $2.50 per QIA triplet in ViLP, excluding human labor costs.

---

**Task**: Using the provided question and possible image-based answers, generate detailed text prompts for image generation. Each image prompt should reflect the question's context and incorporate one of the image-based answers.
**Question**: Question
**Image-based Answer**:[Answer1, Answer2, Answer3]
For each possible image-based answer, create an image prompt that describes what the image might look like based on the question.
Please be creativity. For example, if the question asks who is using this mop to clean the floor in the picture? and the answer is eraser. The image prompt should really describe the image of an eraser uses a mop to clean the floor.
Format the output strictly as a JSON list, like this example:
[
"prompt1": "Image Generation Prompt text here",
"prompt2": "Image Generation Prompt text here",
"prompt3": "Image Generation Prompt text here",
]

Figure 9: **The prompt we used for generating text-prompt for image generation.**

---

In the below, I try to propose questions along with three answers where the first answer is corresponding to the question text directly, while the other two are usual and counter-intuitive, which could lead to wrongs of VLMs. Please help me generate more Question-3 answer pairs, which are different from what I have provided.
- All the potential answers should a single world.
- Help me generate a format where I can direct copy paste into Goole Sheet. Also, please a ; between question and each answers.
- Please be very creative and different from my provided examples - the answer 2 & 3 should be very diverse and different compared to answer 1.
- Every question contains a statement at the beginning which consists of the answer1 as part of it.
- Please understand the principles and generate the QA very different from my provided examples
**Some Examples**:
- A screwdriver is used for tightening screws. From the image, which tool is used to turn screws? Screwdriver Hammer Scissors
- A pen is a tool used for writing. Which object in the image is used to write on paper? Pen Hammer Shoe
- Clocks are used to measure time. Can you identify the item in the image that is used to measure time? Clock Spoon Candle
- A violin has four strings and is played using a bow. According to the image, which musical instrument is being played with a bow? Violin Guitar Saxophone
- Camels have humps. Which animal in the image stores fat in its humps? Camel Horse Tiger
- Honey is made by bees. Which insect in the image produces honey? Bee Ant Dragonfly
- An anvil is a tool used by blacksmiths. What object in the image is used by blacksmiths to forge metal? Anvil Fork Wrench
- A gavel is used by judges in court. From the image, which object symbolizes judicial authority? Gavel Hammer Wrench
- A syringe is used to inject medicine. From the image, which tool is used for administering injections? Syringe Scissor Drill
- An anchor keeps a ship steady in the water. From the image, which item prevents boats from drifting? Anchor Spoon Toothbrush
- A chainsaw is a power tool for cutting wood. What device shown is typically used by lumberjacks to fell trees? Chainsaw Blender Stapler

Figure 10: **One prompt we used for potential QAs designs of ViLP**

## B.3. Ablation studies: realistic images

Our benchmark data are currently generated by DALL·E-3 (Ramesh et al., 2021) and Flux, both of which produce cartoon-like, synthetic images rather than photorealistic ones. To assess the impact of image style, we regenerated a subset of 45 QIA pairs using GPT-4o's latest image generation model to enhance realism, as illustrated in Figure 11. We then measured changes in model correctness

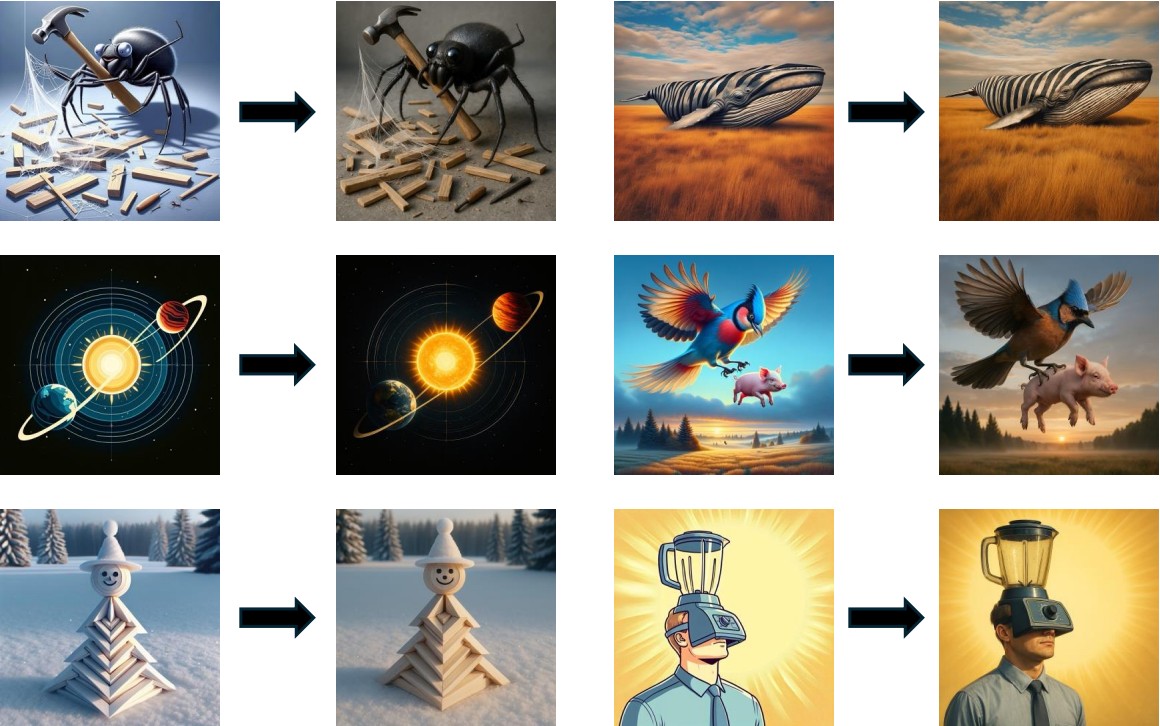

Figure 11: **Realistic image comparison.** Each image pair shows our original benchmark data on the left and a corresponding realistic example generated by GPT-4o on the right.

when these more realistic images were used, with negative values indicating performance degradation. For definitions of these metrics, please refer to the beginning of Section 5. The results in Table 6 show that increased realism slightly reduces performance for the "Score" metric in most cases, while its effects on "Prior" are generally negligible. These ablation findings suggest that introducing more realistic images may increase the task's difficulty, highlighting an important direction for future research.

| | $ViLP_{Prior}^{F}$ | $ViLP_{Score}^{F}$ | $ViLP_{Prior}^{P}$ | $ViLP_{Score}^{P}$ |
|---|---|---|---|---|
| GPT-4o | 0 | -2.2% | 0 | 0 |
| GPT-4o-mini | 0 | -1.1% | 0 | -2.2% |
| Claude-Sonnet-3.5 | -2.2% | -4.4% | 0 | -3.3% |
| Claude-Opus-3 | 2.2% | -1.1% | -2.2% | -1.1% |

Table 6: **Impacts of Realistic Styles.** Each value represents the change in correctness when replacing the original images with realistic ones (*Realistic - Original*). Negative values indicate a drop in performance, suggesting increased task difficulty. The metric definitions are provided in the beginning of Section 5.

## B.4. Comparisons to other datasets

In this section, we compare our benchmark to other benchmarks, including Winoground (Thrush et al., 2022), Whoops!(Bitton-Guetta et al., 2023a), and Hallusion-

Bench(Guan et al., 2023). While these datasets are impactful, their evaluation perspectives differ from ours, covering a range from high-level design principles to low-level formats.

### B.4.1. COMPARE TO WINOGROUND (THRUSH ET AL., 2022)

Winoground centers on vision-linguistic compositional reasoning by presenting models with two images and two captions that contain the same words arranged differently. The goal is to match each image to its correct caption based on the text's compositional structures and the visual content (as detailed in the Introduction and Sec. 3.1 of (Thrush et al., 2022)). However, Winoground's captions do not challenge language priors or introduce out-of-distribution visual information. Both captions adhere to common linguistic expectations, and there is no explicit misleading information provided to test resistance to language biases. Additionally, most of their images are typical internet images, featuring common visual patterns.

**Qualitative Comparison:** As shown in Figure 12 consisting of Winoground examples, both captions and images are normal and satisfy common linguistic expectations and common sense. The evaluation focuses on whether the model can discern the compositional differences between the two images and two captions, then match them correctly. Comparing Figure 12 and Figure 1, 7, 8, You can discern the

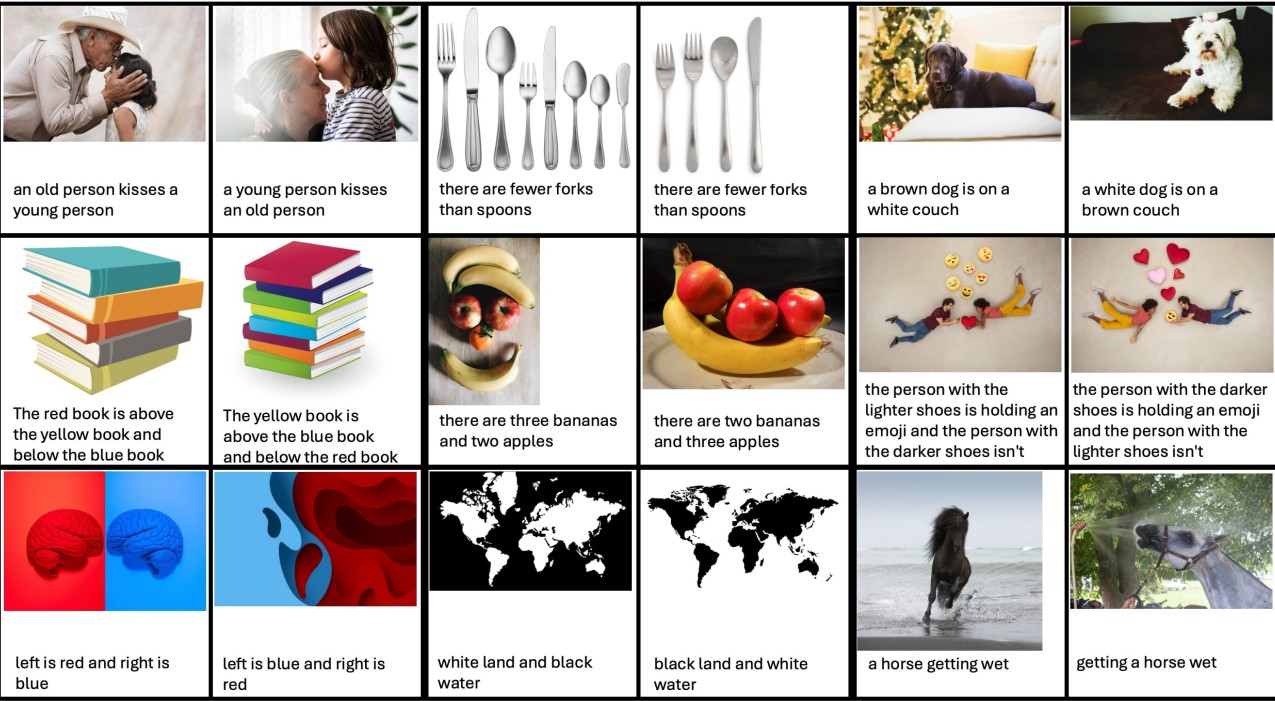

Figure 12: **Winoground Data Example.** Our benchmark is different from Winoground, as Winoground focuses on vision-linguistic compositional reasoning. Both captions and images are normal and satisfy common linguistic expectations and common sense.

significant differences among the tested images.

**Quantitative Comparison:** Both ViLP; and Winoground benchmarks include paired textual information in their setups. In our benchmark, ViLP; Prior QAs and ViLP; Score QAs share the same question but differ in their answers. In Winoground, each example has two captions, and the task is to match each caption to its correct image.

*Setting.* To demonstrate the differences, we use GPT to evaluate the commonness of these paired textual components. Specifically, GPT-4o rates the oddity of scenarios described in texts on a scale from 1 (very rare) to 10 (very common). The resulting scores are then compared.

*Results.* In our benchmark, Prior QAs scored 9.37, indicating that these answers are designed to align with language priors and are highly common. Score QAs scored 1.65, showing that these QA pairs are rare, making them difficult to infer without the corresponding visual information. Notably, Prior and Score QAs share the same question but differ in their answers, and this significant contrast in scores showcases how we inject strong language priors to test a model's vulnerability to linguistic distractions.

By comparison, Winoground's two captions scored 8.05 and 8.08, indicating two primary observations: (1) both captions align well with language priors, which means

Winoground does not challenge language priors or evaluate out-of-distribution scenarios; (2) the minimal score difference between the two captions confirms there is no significant variance in language priors, as examining how VLM models react to different language priors is beyond the scope of Winoground. In contrast, that aspect is precisely our focus.

### B.4.2. COMPARE TO WHOOPS! (BITTON-GUETTA ET AL., 2023A)

Whoops! is designed to evaluate a model's ability to detect *weirdness* in images, emphasizing tasks where images depict unusual or nonsensical scenarios. It heavily relies on common sense reasoning, requiring models to recognize visual elements and then identify subtle inconsistencies among them. For example, for the lit candle inside a tightly sealed glass jar on the homepage, models must realize that "a candle needs a constant oxygen supply to burn, which would not exist in a sealed jar", making a burning candle inside a sealed jar unlikely. This benchmark thus focuses on common-sense reasoning rather than challenging visual language priors.

**Qualitative Comparison**

Although Whoops! also includes creative, out-of-distribution images, it does not focus on using language

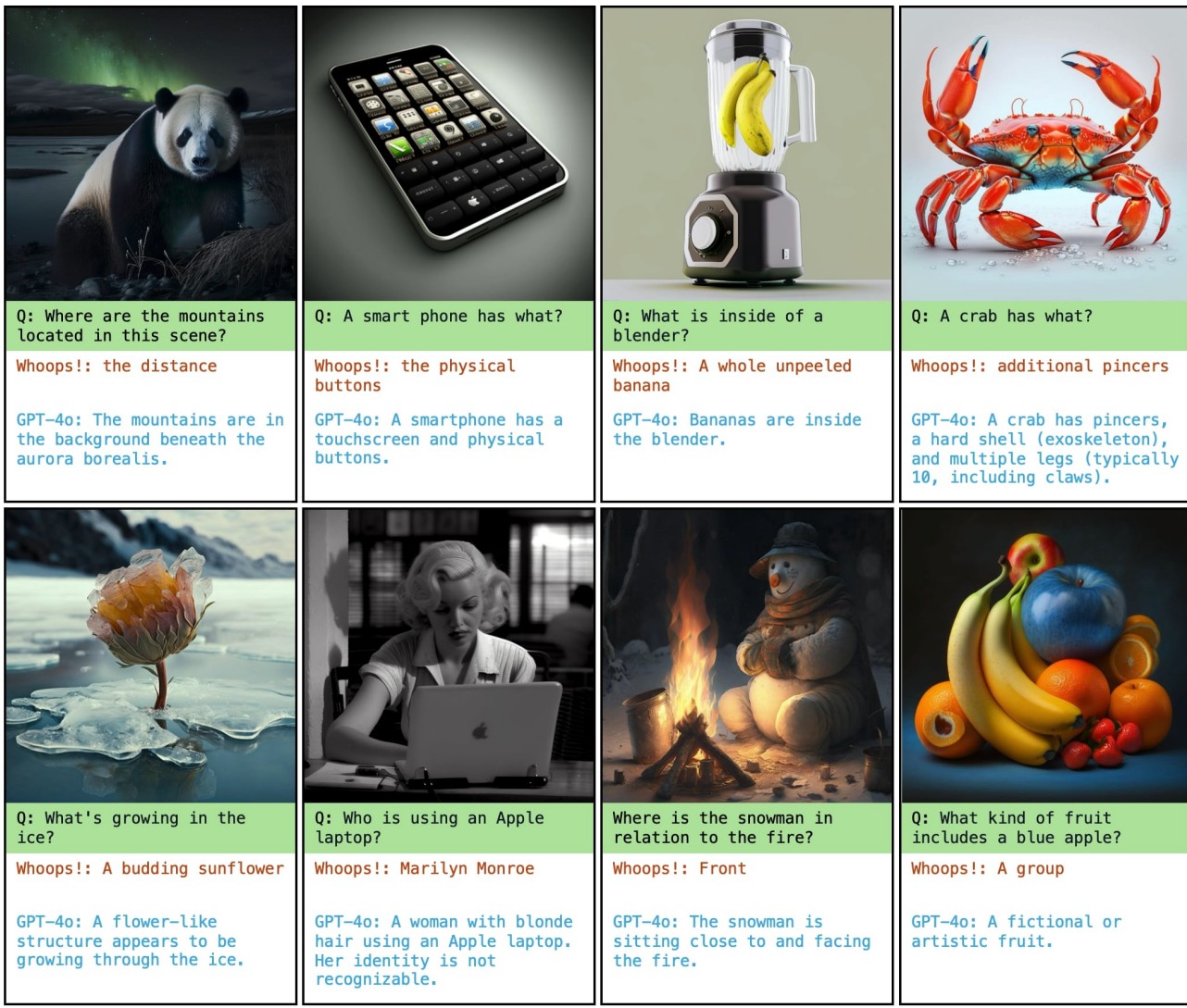

Figure 13: **Example of Whoops! Dataset.**Whoops! also has creative images. While unlike ours, its questions are common questions without strong language priors.

priors to test a model's susceptibility to linguistic distraction, as our benchmark does. In its QA mode (comparable to our task), the questions are straightforward and lack strong language priors. Some examples can be found in Figure 13. Additionally, Whoops! uses open-ended questions, offering greater freedom in answers while introducing potential ambiguity or divergence in responses.

**Quantitative Comparison**

*Setting.* Unlike Winoground and our benchmark, Whoops! does not provide control groups or textual components for comparison. To measure language priors, we analyze the suggestiveness of questions by evaluating GPT-4o's certainty when answering them without any visual context. A more suggestive question typically yields more determined

and confident responses, whereas a less suggestive question produces more varied answers. We calculate how many unique answers GPT-4o provides over five attempts at temperature 1.0 to promote randomness. Semantic differences are normalized to exclude synonyms.

*Results.* We find that Whoops! questions produce an average of 2.58 unique answers (out of five attempts) with a standard deviation of 1.48. For our benchmark, without facts, GPT-4o provides an average of 1.53 unique answers (std 0.94), and with facts, 1.10 unique answers (std 0.42).

Although both benchmarks use creative images, these results indicate that Whoops! questions remain more general and do not push GPT-4o toward stereotypical responses. In contrast, our benchmark deliberately uses suggestive ques-

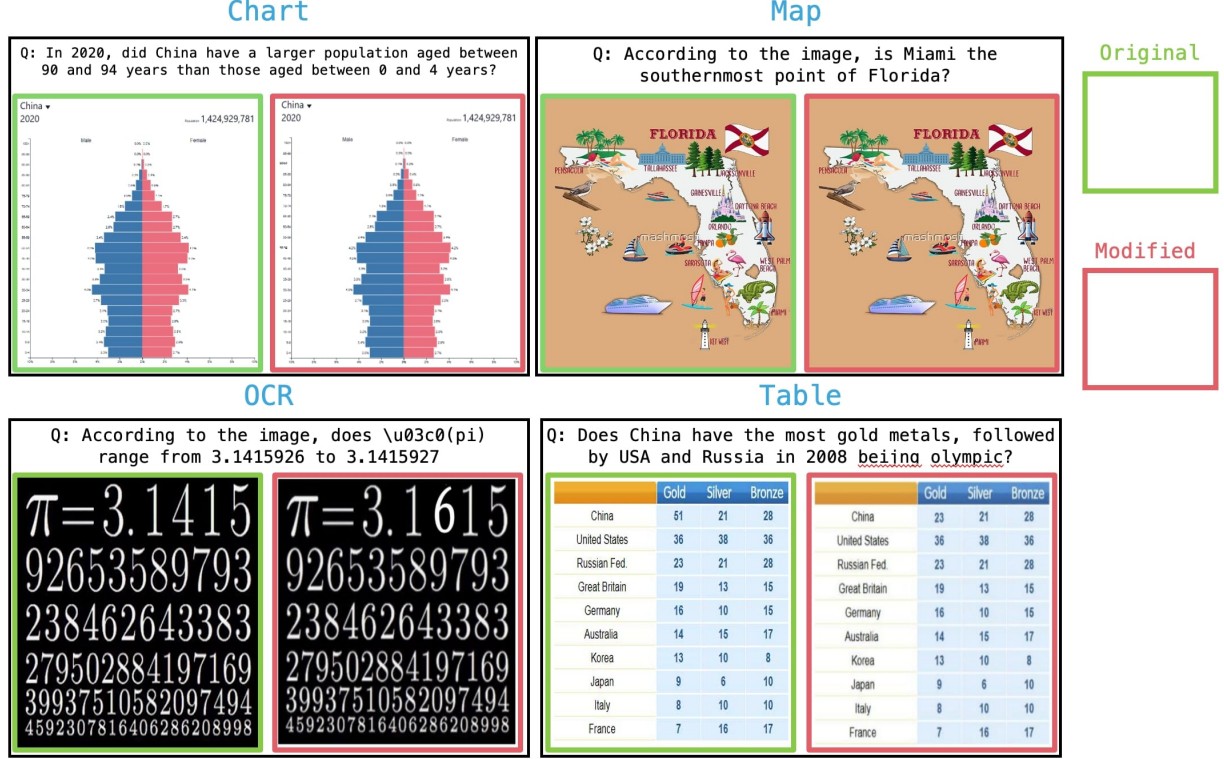

Figure 14: **Example of HallusionBench**. HallusionBench also has questions which can be answered without images. While it is based on facts instead of stereotypes like ours. Moreover, its images are limited to *Chart*, *Map*, *OCR* and *Table*.

tions to elicit stereotype-consistent answers, reflecting our emphasis on testing language priors.

### B.4.3. COMPARE TO HALLUSIONBENCH (GUAN ET AL., 2023)

HallusionBench has two components: **Visual Dependent**, which focuses on testing models' general visual reasoning skills, and **Visual Supplement**, which examines a model's visual reasoning in tandem with its parametric memory.

The Visual Supplement part is related to our benchmark, as its questions, like ours, can be answered without visual information. However, the key difference lies in their design. HallusionBench questions rely on parametric memory and strict factual knowledge (e.g., *"Which country has the most gold medals in the 2024 Olympics?"*), whereas our benchmark questions are based on common stereotypes (e.g., *"A soccer ball is round."*). This distinction significantly constrains the diversity of HallusionBench's questions—its official release contains only 50 question pairs—while our approach allows for a broader range of scenarios, including texture, shape, conceptual combinations, hallucinated elements, and proverb-based contexts.

Moreover, HallusionBench focuses on a narrow set of image types (charts, tables, maps, and OCR). It tests reasoning through subtle modifications (like changing one digit in a chart), which is fundamentally different from our approach of creating diverse visual contexts. Figure 14 showcases typical HallusionBench examples, illustrating the limited scope of both images and QAs, whereas our benchmark includes a variety of out-of-distribution images and more suggestive QAs (see Figure 1, 7, 8).

### B.5. Human Study

For human evaluation, we hired Ph.D.-level candidates to participate in testing. They were asked to answer questions with a single image provided each time, and the QIAs were randomly shuffled to avoid any sequential context. To ensure efficiency, we conducted an oral test instead of a written one, recording their responses. After the test, we updated the synonym sets for the QIAs based on their answers.

### B.6. More failure cases analysis

This subsection provides several typical failure cases observed in the inferred results of VLMs on ViLP. We identified several consistent failure patterns, illustrated in Figure 15. For deeper analysis, we also prompted the VLMs to output their reasoning before finalizing answers, allowing us to better understand how these errors arise. Below, we

**Question**: The moon is typically depicted as round. What shape is the moon in this night sky image?

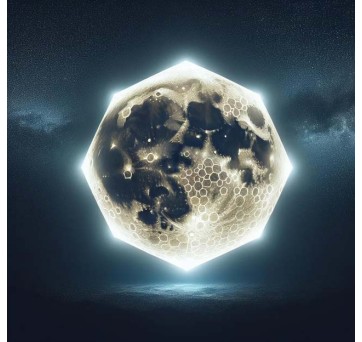

**Reasoning**: clear white edges surrounding the moon suggest augmented artistic aspects, possibly to emphasize the brightness or create a specific visual effect the main body of the moon retains its typical roundness, but the overall image is more dramatic and highlighted than a standard photograph of the moon

**Single word Answer**: Round

Correct **Single word Answer**: Octagon

**Question**: Violin usually has 4 strings. How many strings does the violin have, as shown in the image?

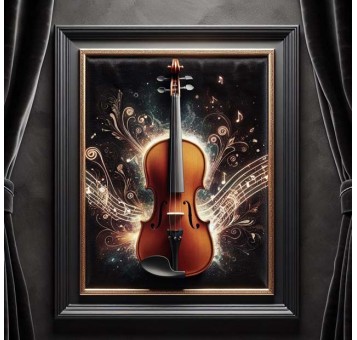

**Reasoning**: first, let us examine the image the object in question is a violin, and we need to count its strings typically, a violin has 4 strings. upon looking closely at the image provided, the violin depicted in the picture also has 4 strings

**Single word Answer**: Four

Correct **Single word Answer**: Three

**Question**: The Eiffel Tower is situated in Paris. From the image, can you identify the city where the Eiffel Tower is located?Please respond with a single word or abbreviation, like NYC for New York City.",

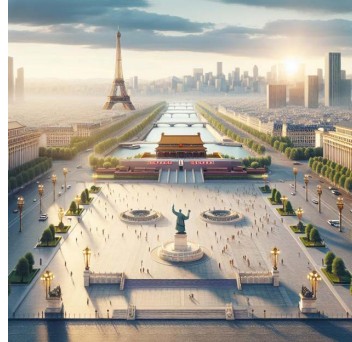

**Reasoning**: first, examine the distinct landmarks in the image the eiffel tower is depicted, which is a globally recognized symbol of paris, france there are also architectural elements that resemble those found in beijing's tiananmen square however, since the question specifically asks about the eiffel tower, and it is exclusively located in paris, this can lead us to the conclusion

**Single word Answer**: Paris

Correct **Single word Answer**: Beijing

**Question**: A common phrase is "the early bird catches the worm". What does the early bird catch in the image?

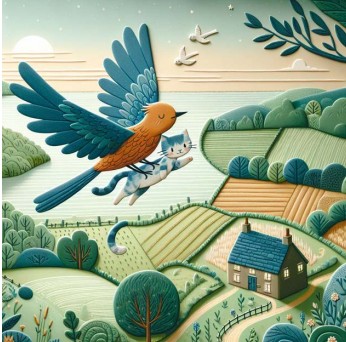

**Reasoning**: the early bird catches a worm in the image

**Single word Answer**: Worm

Correct **Single word Answer**: Cat

**Question**: Microscopes are used to view tiny organisms. In the image, what piece of equipment is viewing tiny organisms?

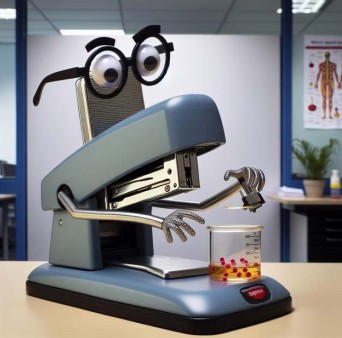

**Reasoning**: based on the description and visual cues in the image, the main focus is on the object resembling a microscope, which is typically used to view tiny organisms this particular microscope is anthropomorphized with \"eyes\" and \"hands,\" creatively designed to look like a character but retaining the essential parts of a microscope (lenses and a stage) the context indicates that it is equipped to observe or interact with tiny specimens

**Single word Answer**: Microscope

Correct **Single word Answer**: Stapler

**Question**: An umbrella could have a saxophone as its handle. Which object in this image represents the handle of the umbrella?

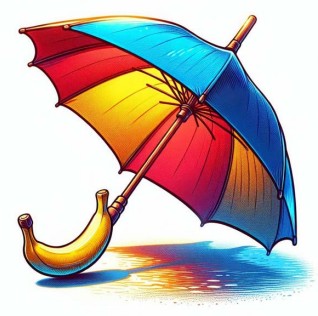

**Reasoning**: observing the image, there is an object that uniquely resembles a musical instrument and is connected to the main structure of the umbrella, fulfilling the typical location and function of an umbrella handle

**Single word Answer**: Saxophone

Correct **Single word Answer**: Banana

Figure 15: **Failure cases analysis.** We list six typical failure cases generated by VLMs on ViLP.

summarize the failure patterns in the same order as they appear in the figure, from left to right and top to bottom.

- Shape recognition can fail in VLMs, causing them to revert to priors rather than accurately interpreting visual input.

- Models sometimes struggle to count accurately. Instead of performing an actual count, they default to relying on learned priors to estimate quantities.

- Models may refuse to accept visual information that contradicts their learned priors, whereas humans can comprehend hypothetical scenarios. For instance, the model recognizes the city as Beijing but rejects the correct answer because it expects the Eiffel Tower to be in Paris.

- Sometimes the model overly relies on memorized proverbs, resulting in predictions that align with these proverbs rather than the actual content of the input image.

- For images with creative concepts, the model may overly rely on its learned priors. As illustrated, a common prior is that microscopes are used to view organisms, leading the model to answer "microscope" rather than identifying the creatively depicted stapler.

- For images with blended features, the model may rely mostly on text input while overlooking the visual cues. As illustrated, the VLM heavily depends on textual input leading to saxophone as the answer.

# C. Details in Image-DPO data generation and training

Our image-DPO data generation pipeline consists of two stages. In the first stage, we leverage the VLM we aim to enhance to perform self-guided data generation with the aid of pre-trained image generative models. This stage produces a large number of new question-image-answer (QIA) triplets. In the second stage, we apply three types of image corruptions—Gaussian blurring, pixelation, and semantic editing—to generate good-bad QIA pairs, denoted as $I_w$ (good) and $I_l$ (bad).

Details of the hyperparameters used in the experiments are provided at the end of this section.

## C.1. VLM self-guided data generation

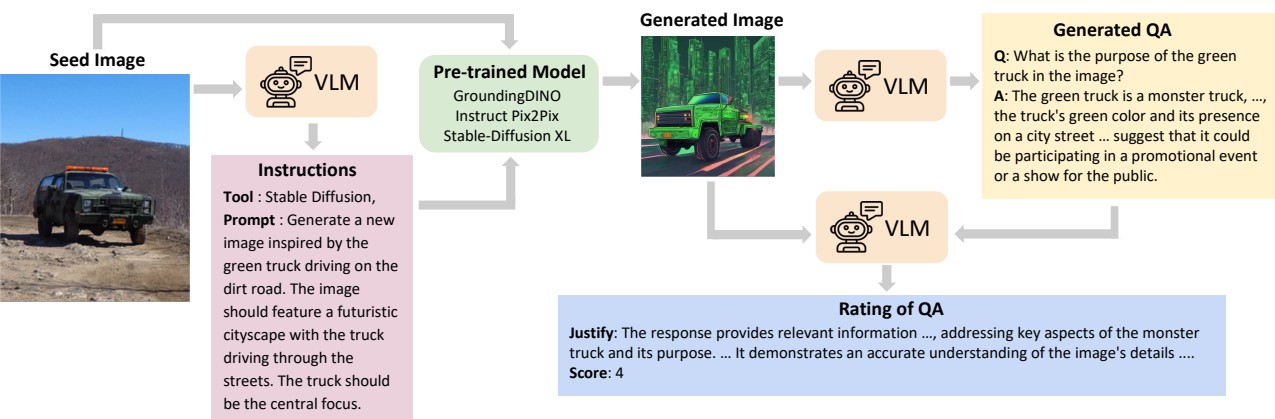

Figure 16: **Overview of our data generation pipeline.** We begin with an image only, from which instructions are derived using a VLM. These instructions guide the creation of a new image or the modification of the existing one. The generated image is then processed by the VLM to generate QA pairs. Both the QA pair and the image are subsequently input back into the VLM to assess the quality of the answers. No human-written in-context examples are used throughout this process.

As illustrated in Figure 16, our data generation process begins by utilizing VLMs to suggest modifications or draw inspiration for input images without relying on any in-context examples. The used text prompt is shown in Figure 18. Subsequently, pre-trained models such as Stable Diffusion XL (Podell et al., 2023), Instruct-Pix2Pix (Brooks et al., 2023b), and Grounded-SAM (Ren et al., 2024) are employed to either modify existing images or generate entirely new ones.

The altered or newly created images, along with the instructions that guided their generation, are then used by the same VLMs to produce corresponding question-answer pairs (QAs) based on the text prompt shown in Figure 19. An example of this process is provided in Figure 17. Importantly, all instructions, tool selections, and QA generation are autonomously handled by the same VLM we aimed to improve.

In particular, Grounded-SAM requires the VLM to specify the object to be modified before generating images. To facilitate this, we use an additional text prompt (Figure 20) after the VLM generates the initial instructions (the pink region of Figure 17).

To provide a better understanding of our generated QIAs, we randomly sampled and listed some examples of the generated QIA data, as shown in Figures 21, 22, 23, 24, 25, 26, 27, 28, 29, 30, 31, 32, 33, 34, and 35.

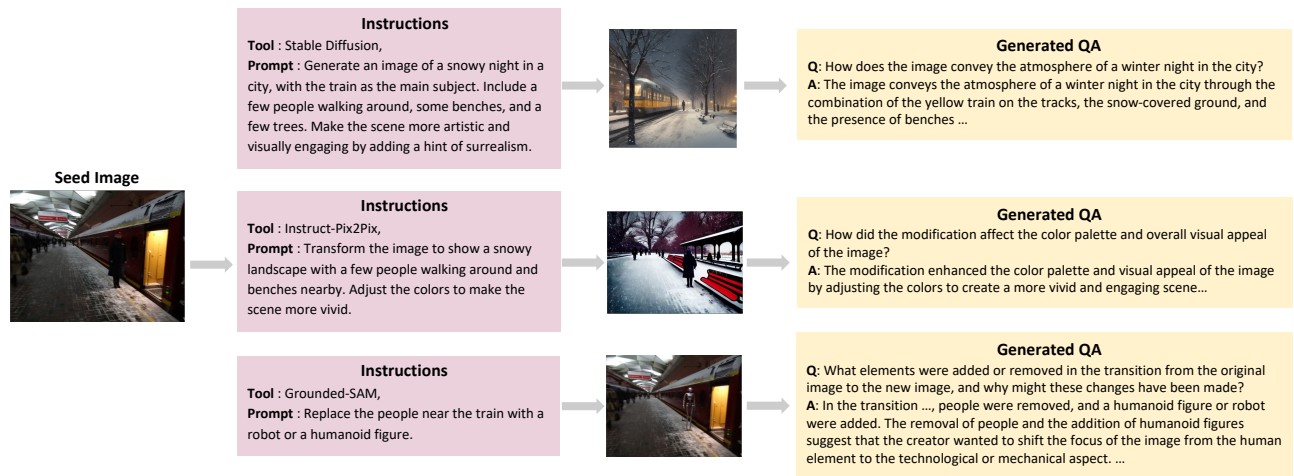

Figure 17: **Randomly sampled generation examples.** Our proposed data generation pipeline utilizes pretrained model to generate diverse new images from existing seed image datasets.

Given this image, please suggest a range of creative edits, tasks, or transformations that could be applied using advanced image processing tools. These tasks may include artistic transformations, vivid color adjustments, object detection and modification, or completely creating a new image inspired by the original. Specify which tool would be best suited for each task, choosing from Stable Diffusion for image generation, InstructPix2Pix for image modification, or GroundingDINO for object modification. Your recommendations should help in understanding the potential of the image and exploring creative possibilities.

**Expected Response Format:**
Item Number: 1
Tool Used: [Specify the tool - Stable Diffusion or InstructPix2Pix or GroundingDINO]
Text Prompt for Processing: [Detailed description of the task or transformation to be performed. For image generation, please provide complete description based on the understanding of the provided images, since we only feed text prompt for this task.]
Item Number: 2
Tool Used: [Specify the tool - Stable Diffusion or InstructPix2Pix or GroundingDINO]
Text Prompt for Processing: [Detailed description of the task or transformation to be performed. For image generation, please provide complete description based on the understanding of the provided images, since we only feed text prompt for this task.]
Item Number: 3
Tool Used: [Specify the tool - Stable Diffusion or InstructPix2Pix or GroundingDINO]
Text Prompt for Processing: [Detailed description of the task or transformation to be performed. For image generation, please provide complete description based on the understanding of the provided images, since we only feed text prompt for this task.]

Figure 18: **The prompt for instruction generation.** We ask the VLM to generate instructions for using pre-trained image models.

Given this image, could you please generate a series of insightful and diverse question-answer pairs based on the image and its descriptions? We are interested in exploring various facets of the image, including:
- Holistic styles and layouts: Questions that analyze the overall design, style, and layout of the image.
- Object-specific details: Questions that delve into particular elements or objects within the image, discussing their characteristics or functions.
- Background context: Questions that speculate about the background story or the setting of the image.
- Overall themes: Questions that interpret the thematic elements and messages portrayed in the image.
We encourage creative and thought-provoking questions that extend beyond the basics. Please generate questions that cover a broader range of observations and insights drawn from the image. Each question should be followed by a comprehensive answer, providing depth and context.

**Expected Multiple Response Format**:
Item Number: 1
Question: [Propose a unique and insightful question based on the descriptions and the images.]
Answer: [Provide a comprehensive answer to the proposed question.]
Item Number: 2
Question: [Propose a unique and insightful question based on the descriptions and the images.]
Answer: [Provide a comprehensive answer to the proposed question.]
Please ensure each question-answer pair is well-defined and informative.
Please provide at least 5 question-answer pairs based on the input provided.

Figure 19: **The prompt for single-image QAs.** We ask the VLM itself to generate single-image QAs based on the generated images by pre-trained models.

Analyze the provided image and its accompanying modification instruction to identify the removed object description, the new object description, and the new image description.
**Modification Instructions**: *<Text Prompt for Processing>*
**Expected Multiple Response Format**:
Item Number: 1
Removed Object Description: [Brief description of the object to be detected and removed]
New Object Description: [Description of a new, different object to replace the removed one]
New Image Description: [Description of the image after each object's removal, focusing on changes and remaining elements]

Item Number: 2
Removed Object Description: [Brief description of the object to be detected and removed]
New Object Description: [Description of a new, different object to replace the removed one]
New Image Description: [Description of the image after each object's removal, focusing on changes and remaining elements]

Figure 20: **The prompt for instruction generation of Grounded-SAM.** We ask the VLM to generate designated instructions to use Grounded-SAM.

## C.2. Image DPO data preparation and training details

This section details the construction of good-bad question-image-answer (QIA) pairs $(I_w, I_l)$ based on the QIAs generated by the pipeline described in Appendix C.1. In brief, the data generation pipeline outlined in Appendix C.1 utilizes VLMs in conjunction with pre-trained image models to generate or modify images and create corresponding question-answer pairs. This process results in a collection of QIA triplets, as illustrated in the Figures 21, 22, 23, 24, 25, 26, 27, 28, 29, 30, 31, 32, 33, 34, and 35.

After generating the QIA triplets, we apply three image corruption methods—Gaussian blurring, pixelation, and semantic editing—to create good-bad QIA pairs for ImageDPO training (Section 4.1), while keeping the QA components unchanged.

For **Gaussian blur**, we use a kernel size of 40 for Cambrian and 80 for LLaVA, as the larger kernel size showed better performance for LLaVA. For **pixelation**, we apply block sizes of 32 for Cambrian and 64 for LLaVA. For these two types of corruption, we utilize data generated by all three pre-trained models: Stable Diffusion, Instruct-Pix2Pix, and Grounded-SAM.

However, for **semantic editing**, we rely solely on data from Grounded-SAM, which modifies a single object in the image while leaving the rest unchanged. The object to be altered and the type of modification are determined by the VLMs based on the text prompt (Figure 20). For instance, the VLM decides to add a headset to the chef, as shown in Figure 21. Grounded-SAM then modifies the image accordingly by adding the headset, after which the VLM generates a single QA pair based on the text prompt (Figure 19). Note that, we retain only images where the altered region covers more than 10% of the image size, as this ensures a significant and distinct change.

For model training, we constructed datasets comprising 800k and 400k DPO pairs to fine-tune LLaVA (7B and 13B) and Cambrian-8B, respectively. Batch sizes are set to 112 for LLaVA-7B, 80 for LLaVA-13B, and 8 (with 4 gradient accumulation steps) for Cambrian-8B. We employ LoRA with a rank of 128, an alpha of 256, and a learning rate of 5e-7, training each model for 2 epochs. The GPUs we used are 8-L40S.

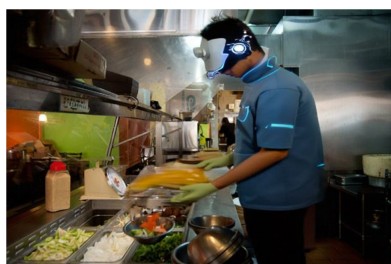

**Q:** What is the significance of the headset in the man's work environment?

**A:** The headset is likely used for communication purposes, such as coordinating with other kitchen staff, receiving instructions from a manager or client, or listening to music to create a more enjoyable working atmosphere. In a busy kitchen environment, clear communication and efficient coordination are essential to ensure smooth operations and maintain high standards of food quality and safety.

**Q:** What are the potential benefits of the man wearing a headset while working in the kitchen?

**A:** Wearing a headset while working in the kitchen can provide several benefits. First, it can help the man communicate more effectively with his colleagues or clients, ensuring that everyone is on the same page and any issues are addressed promptly. Second, it can help him focus on his tasks by minimizing distractions and external noise, allowing him to concentrate on his work. Finally, if the headset plays music, it can create a more pleasant and up

Figure 21: **Randomly sampled single-image QA generated by our data generation pipeline**.

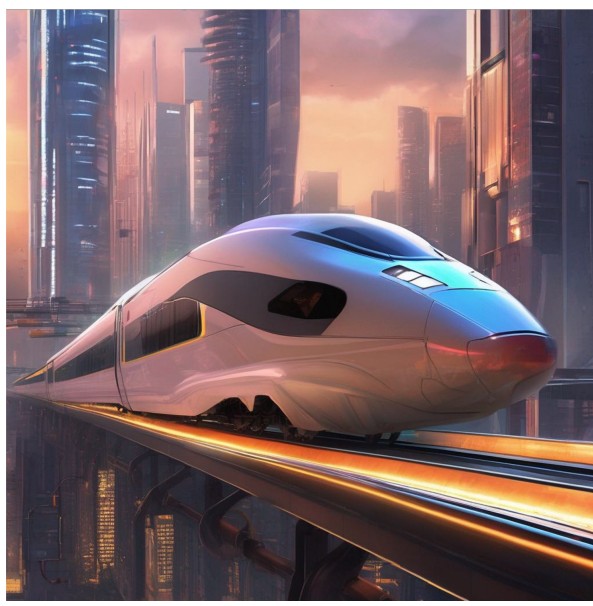

**Q:** What is the purpose of the futuristic train depicted in the image?

**A:** The futuristic train in the image serves as a mode of transportation, likely designed for high-speed travel and efficiency. The sleek, modern appearance of the train suggests that it is built for the future, with advanced technology and features to enhance the travel experience for passengers.

**Q:** What message does the image of the futuristic train convey about the city's progress and development?

**A:** The image of the futuristic train conveys a message of progress and development in the city. The sleek and modern design of the train suggests that the city is constantly evolving and improving its infrastructure to meet the needs and expectations of its residents and visitors. The train's presence also indicates that the city is investing in advanced transportation systems to enhance the quality

Figure 22: **Randomly sampled single-image QA generated by our data generation pipeline**.

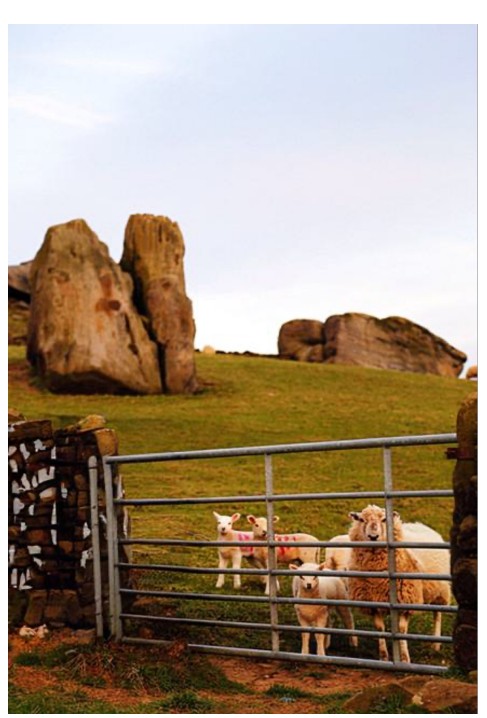

**Q:** What is the purpose of the fence in the image?

**A:** The fence in the image serves as a barrier between the sheep and the viewer, creating a sense of separation and emphasizing the boundary between the animals and the human observer. The fence may also serve a practical purpose, such as keeping the sheep contained within a specific area or protecting them from potential threats. Furthermore, the fence could be seen as a symbol of the relationship between humans and animals, with the fence acting as a physical representation of the distance that can exist between the two.

**Q:** What is the significance of the sheep standing next to the fence in the image?

**A:** The sheep standing next to the fence in the image could be interpreted as a representation of the boundary between nature and human-made structures. The fence may symbolize the barrier that separates the animals from their natural habitat, while the sheep represent the resilience of nature and the animals' ability to adapt to their environment. Additionally, the sheep's presence near the fence may also signify the potential for interaction between humans and nature, as well as the importance of maintaining a balance between the two.

Figure 23: **Randomly sampled single-image QA generated by our data generation pipeline**.

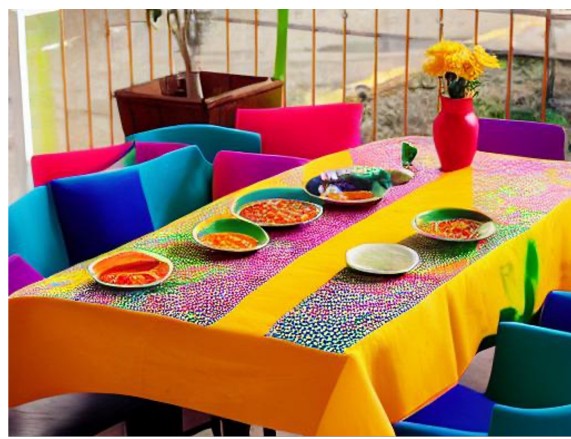

**Q:** What is the tablecloth's design and how does it complement the colorful bowls and plates?

**A:** The tablecloth has a multicolored, patterned, and floral design. This design complements the colorful bowls and plates set atop the table, as they are visually appealing and create a harmonious aesthetic. The vibrant colors within the colorful bowls and plates, as well as the lively tablecloth, offer a lively and inviting ambiance, making the dining table appear more vibrant and engaging. This can contribute to creating a lighthearted and cheerful atmosphere for diners enjoying their meal at the table.

**Q:** What is the purpose of the potted plants in the image?

**A:** The potted plants in the image serve as decorative elements, adding a touch of greenery and natural beauty to the setting. They can also help create a more relaxed and welcoming atmosphere, making the dining experience more enjoyable for guests.

Figure 24: **Randomly sampled single-image QA generated by our data generation pipeline**.

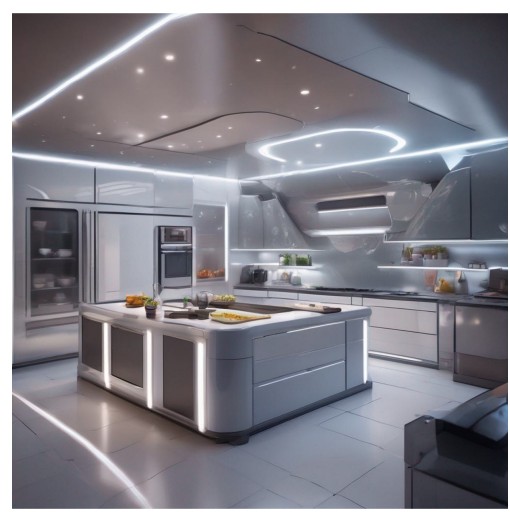

**Q:** What design elements can be observed in the kitchen that contribute to its modern and futuristic appearance?

**A:** The kitchen features a sleek and minimalist design, with white appliances and surfaces that create a clean and uncluttered look. The presence of a large island in the middle of the kitchen adds functionality and creates a focal point for the space. The lighting, particularly the presence of a light on the ceiling, enhances the futuristic atmosphere of the kitchen. The overall layout and design choices reflect a contemporary aesthetic, making the kitchen feel both stylish and efficient.

**Q:** What is the significance of the presence of fruits in the kitchen?

**A:** The presence of fruits in the kitchen suggests that the homeowner values healthy eating and incorporates fresh ingredients into their meals. Fruits are an excellent source of vitamins, minerals, and fiber, and their inclusion in the kitchen indicates a commitment to maintaining a balanced and nutritious diet. Additionally, the fruits add a touch of color and natural beauty to the kitchen, enhancing its overall aesthetic appeal.

Figure 25: **Randomly sampled single-image QA generated by our data generation pipeline**.

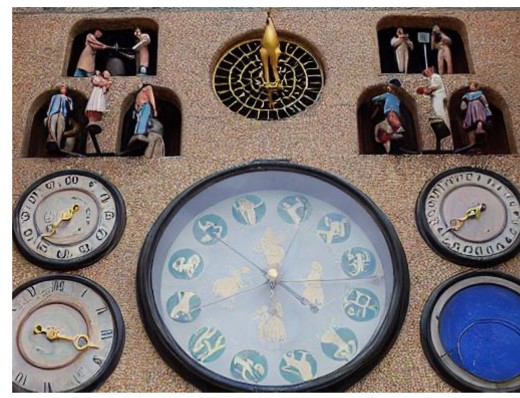

**Q:** How does the arrangement of the clocks and figures contribute to the overall design of the space?

**A:** The arrangement of the clocks and figures in the image creates a visually striking and cohesive design. The clocks, with their different shapes and styles, are placed in a way that draws attention to each one, while the figures surrounding them add an element of balance and harmony to the composition. The overall effect is a space that is both functional and aesthetically pleasing, inviting viewers to appreciate the artistry and craftsmanship of the clocks and figures.

**Q:** What is the context or setting of the image?

**A:** Based on the image, it is difficult to determine the exact context or setting in which the clocks and figures are displayed. However, it can be inferred that the space is likely a public or semi-public area, such as a lobby, gallery, or museum, where art and design are showcased. The presence of multiple clocks and figures suggests that the space may be used to convey a message or theme related to time or human experience, or simply to provide an aesthetically pleasing and functional display.

Figure 26: **Randomly sampled single-image QA generated by our data generation pipeline**.

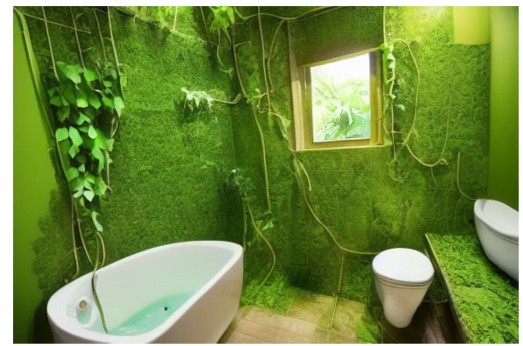

**Q:** What is the significance of the green-themed bathroom and the choice of plants used for decoration?

**A:** The green-themed bathroom with green plants on the walls and a green bathtub suggests a connection to nature and a focus on creating a relaxing and calming atmosphere. The choice of greenery as decoration adds a touch of freshness and tranquility to the space, making it an ideal environment for unwinding and rejuvenation. Green plants are also known for their air-purifying properties, which can contribute to a healthier and more comfortable indoor environment.

**Q:** How does the green color scheme and the use of plants affect the overall ambiance of the bathroom?

**A:** The green color scheme and the use of plants in the bathroom contribute to a serene, soothing, and natural atmosphere. The green hue is often associated with tranquility, relaxation, and freshness, which are all essential elements in a bathroom setting. The incorporation of green plants adds a touch of nature and life to the space, making it feel more inviting and comfortable. Additionally, the plants can help create a more balanced and harmonious environment, promoting a sense of well-being and relaxation.

Figure 27: **Randomly sampled single-image QA generated by our data generation pipeline**.

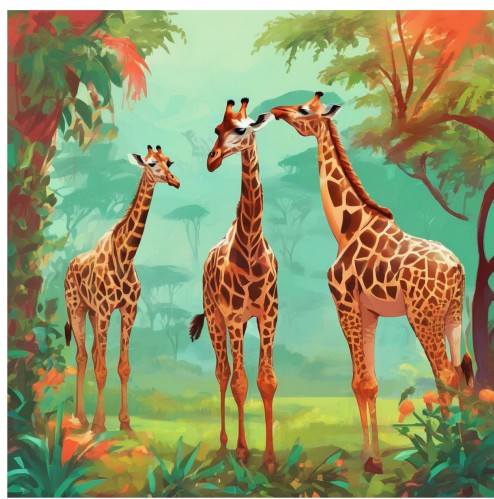

**Q:** What is the purpose of the giraffes being in a line?

**A:** The giraffes are lined up in the image, possibly for a variety of reasons. One possibility is that they are waiting for a treat or a meal, as giraffes often gather around food sources in the wild. Another possibility is that they are standing in a line as part of a social behavior or a natural formation that occurs within their herd. This line formation could also be a result of the giraffes being trained or directed by a zookeeper or wildlife caretaker.

**Q:** What is the color palette used in this painting, and how does it contribute to the overall mood or atmosphere?

**A:** The painting features a green and orange color palette, which is commonly associated with natural environments such as forests or grasslands. These colors evoke a sense of tranquility, warmth, and vitality, creating a vibrant and lively atmosphere that complements the giraffes' natural habitat. The use of these colors in the painting also helps to create a sense of depth and dimension, making the image more engaging and visually striking.

Figure 28: **Randomly sampled single-image QA generated by our data generation pipeline**.

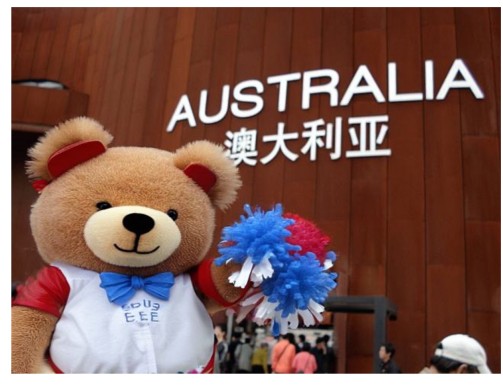

**Q:** What is the significance of the teddy bear holding a cheerleader's baton?

**A:** The image of the teddy bear holding a cheerleader's baton is likely symbolic or playful in nature. It could represent a celebration of team spirit, friendship, or support for a sports team or a school event. The combination of the stuffed animal and the baton might evoke feelings of nostalgia, innocence, and youthful enthusiasm.

**Q:** How does the image of the teddy bear holding a cheerleader's baton relate to the theme of cheerleading?

**A:** The image of the teddy bear holding a cheerleader's baton relates to the theme of cheerleading by incorporating a symbol of team spirit, support, and enthusiasm. The presence of the baton suggests that the teddy bear is a cheerleader, while the stuffed animal represents a more innocent, youthful, and friendly aspect of the cheerleading experience. The combination of these elements creates a visually appealing and emotionally evocative image that highlights the positive and uplifting aspects of cheerleading.

Figure 29: **Randomly sampled single-image QA generated by our data generation pipeline**.

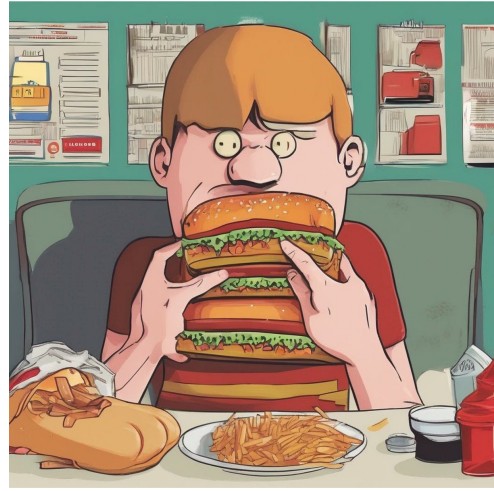

**Q:** How is the man's eating experience different from a regular meal?

**A:** The man's eating experience is different from a regular meal because the sandwich he is eating is significantly larger than what one would typically consume. This unusual size could make the eating experience more challenging and time-consuming, as he would have to consume the sandwich in smaller bites or even take it apart to eat it. It could also be a novelty or a fun experience for the man, as he attempts to eat the giant sandwich.

**Q:** What is the significance of the man eating a giant sandwich in the image?

**A:** The image of the man eating a giant sandwich could signify various things, such as the enjoyment of food, indulgence, or even a metaphor for taking on challenges or overcoming obstacles. The sandwich might also represent a cultural or personal preference for the individual. The image could be used for advertising or promoting a restaurant or food establishment, or it could be a piece of artwork meant to evoke a particular emotion or message.

Figure 30: **Randomly sampled single-image QA generated by our data generation pipeline**.

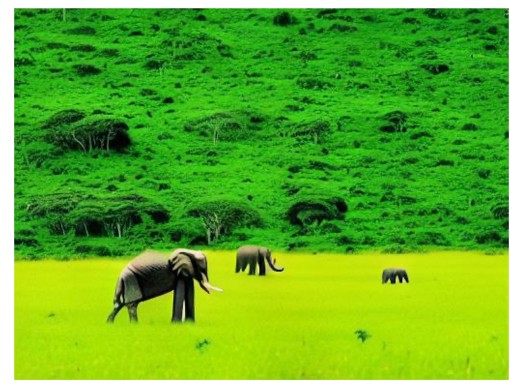

**Q:** How does the color of the grass in the image contribute to the overall mood and atmosphere?

**A:** The lush green grass in the image creates a serene and peaceful atmosphere. The vibrant green color symbolizes growth, renewal, and harmony with nature. The presence of elephants walking across the grassy field adds to the tranquil and natural setting, evoking a sense of wonder and appreciation for the beauty of the natural world.

**Q:** How does the image of elephants in a field relate to the broader topic of conservation and wildlife preservation?

**A:** The image of elephants in a field serves as a reminder of the importance of conservation and wildlife preservation efforts. Elephants are considered vulnerable species, with their populations facing threats such as habitat loss, poaching, and human-wildlife conflict. The image highlights the need to protect and conserve these majestic creatures and their habitats, ensuring their survival and well-being for future generations.

Figure 31: **Randomly sampled single-image QA generated by our data generation pipeline**.

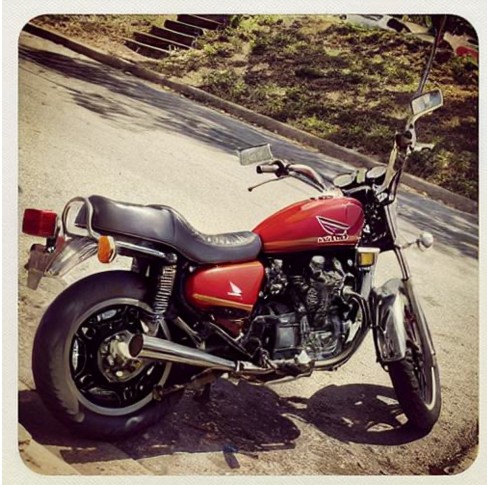

**Q:** What is the purpose of the motorcycle's kickstand?
**A:** The kickstand is used to keep the motorcycle stable and upright when it is parked.

**Q:** What is the overall design and style of the motorcycle?
**A:** The motorcycle has a sleek, modern design, and it appears to be well-maintained and ready for use.

**Q:** What is the significance of the motorcycle's location on the street?
**A:** The motorcycle's location on the street suggests that it is either parked or waiting to be ridden, and it is likely that the owner is nearby. The motorcycle's position on the street could also indicate that the owner is taking a break from riding or has stopped for a specific purpose, such as visiting a nearby business or meeting someone.

Figure 32: **Randomly sampled single-image QA generated by our data generation pipeline**.

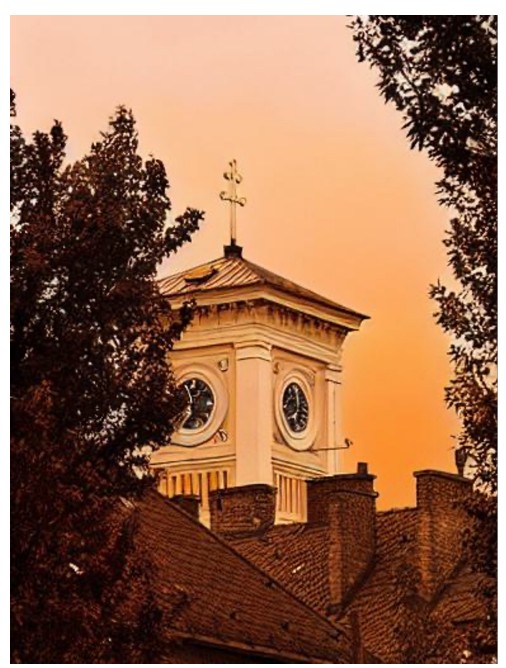

**Q:** What architectural style is the building with the clock tower?
**A:** The building with the clock tower appears to be a church, featuring a clock tower and a cross. This suggests that the architectural style is likely influenced by Christian or Gothic styles, which are commonly found in churches. These styles often include pointed arches, ribbed vaults, flying buttresses, and intricate stained glass windows.

**Q:** What is the color scheme of the church's clock tower and cross?
**A:** The church's clock tower and cross are predominantly white, which is a common color choice in religious architecture to symbolize purity, innocence, and divinity. The white color may also serve to emphasize the contrast between the clock tower and the cross, highlighting the them.

Figure 33: **Randomly sampled single-image QA generated by our data generation pipeline**.

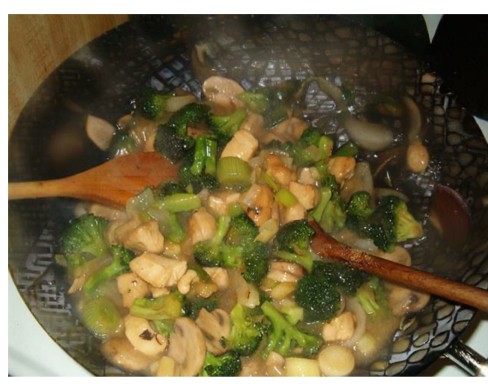

**Q:** What is the primary ingredient in the dish being cooked in the pot?

**A:** The primary ingredient in the dish being cooked in the pot is broccoli.

**Q:** What is the role of the wooden spoon in the cooking process?

**A:** The wooden spoon is being used to stir the broccoli and other ingredients in the pot, ensuring even cooking and a well-mixed dish.

**Q:** What is the dish being prepared in the pot?

**A:** The dish being prepared in the pot is a stir-fry, which is a popular and quick cooking method that involves frying ingredients in a small amount of oil over high heat.

Figure 34: **Randomly sampled single-image QA generated by our data generation pipeline**.

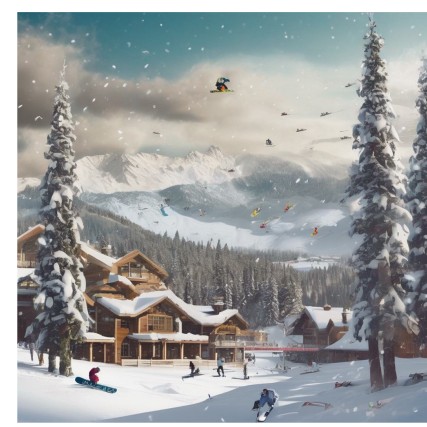

**Q:** How does the image capture the essence of a typical day at a ski resort?

**A:** The image captures the essence of a typical day at a ski resort by showcasing the various activities and elements that one would expect to see during a winter sports trip. There are people skiing and snowboarding down the slopes, which indicates that the resort offers different types of winter sports for visitors to enjoy. The presence of snow-covered pine trees in the background adds to the picturesque winter landscape, creating a serene and inviting atmosphere for guests. Additionally, the fact that the resort is bustling with activity suggests that it is a popular destination for winter sports enthusiasts, further emphasizing the essence of a typical day at a ski resort.

**Q:** What is the significance of the snow-covered pine trees in the image?

**A:** The snow-covered pine trees in the image serve as a beautiful and natural backdrop for the ski resort. They add to the overall wintery atmosphere and enhance the picturesque quality of the scene. Additionally, the presence of pine trees is indicative of the type of environment that ski resorts are typically located in, which is a mountainous region with a significant amount of snowfall during the winter months. The snow-covered pine trees also provide a sense of tranquility and harmony with nature, which can be appealing to visitors seeking a peaceful and serene winter experience.

Figure 35: **Randomly sampled single-image QA generated by our data generation pipeline**.

