# OpenReview forum: "Probing Visual Language Priors in VLMs"
_ICML.cc/2025/Conference — ICML 2025 poster_

### Official Review · Reviewer_hkyN · 2025-03-12

**Overall Recommendation:** 3

**Summary:**

This paper investigates visual language priors in vision-language models , analyzing how these models rely on textual biases rather than true visual reasoning. The authors introduce ViLP, a benchmark designed to expose such priors by presenting models with out-of-distribution  images and distractor text-based facts. The benchmark consists of question-image-answer triplets, where the correct answer contradicts common textual priors, requiring models to focus on visual reasoning.

To address the issue of over-reliance on text priors, the paper proposes Image-DPO, a training framework that introduces controlled image corruptions to force VLMs to develop stronger visual grounding. Experiments with LLaVA-v1.5 and Cambrian demonstrate that Image-DPO improves performance on ViLP and other multimodal benchmarks.

**Claims And Evidence:**

Yes

**Essential References Not Discussed:**

See weakness.

**Experimental Designs Or Analyses:**

See weakness.

**Methods And Evaluation Criteria:**

See Weakness.

**Other Comments Or Suggestions:**

See weakness.

**Other Strengths And Weaknesses:**

### **Strengths**
- An interesting benchmark filling a gap in VLM research.
- The proposed Image-DPO method improves model performance.
- Comprehensive evaluation across multiple models.

---

### **Weaknesses**
- **Lack of in-depth failure analysis.**
  - A more detailed breakdown of failure cases would be beneficial. For instance, what are the top three most common failure types in this benchmark?

- **Methodological concerns regarding Image-DPO.**
  - Weak connection between Image-DPO and the benchmark itself. The proposed method does not seem inherently tied to addressing language priors, making Section 4 feel somewhat disconnected from the core benchmark motivation.
  - Lack of distinction from prior DPO methods. The paper does not clearly differentiate Image-DPO from existing DPO approaches, such as those in [1], which also augment images.

- **Benchmark realism and case selection.**
  - Some examples in the benchmark (e.g., Figure 3 with unnatural tiger and moon images) appear unrealistic.
  - Comparatively, benchmarks like Winoground and HallusionBench provide more realistic settings with real-world images and practical vision-language challenges.
  - A better approach might be to incorporate real images using image retrieval techniques, such as:
    - Identifying common real-world priors (e.g., cows grazing on grass)
    - Introducing subtle variations (e.g., a rabbit on the grass instead of a cow)
    - This approach aligns well with the paper’s motivation of probing visual priors.

- **Insufficient explanation of LLaVA-OneVision performance in Table 2.**
  - The paper does not provide a clear justification for LLaVA-OneVision’s performance in Table 2.
  - If the evaluation method penalizes long responses where the correct answer is embedded within a longer explanation, this could raise concerns about the robustness of the evaluation criteria.
  - Could you provide failure case of this model?

---

### **Reference**
[1] *Self-Supervised Visual Preference Alignment.*

**Questions For Authors:**

See weakness.

**Relation To Broader Scientific Literature:**

Related to prior work on textual priors in VQA datasets (e.g., Winoground, HallusionBench).

**Theoretical Claims:**

Yes. The paper provides a theoretical connection between Image-DPO and RLHF objectives.

---

> ### Author Rebuttal · Authors · 2025-04-01
>
> We greatly appreciate the detailed and insightful feedback. Below, we have carefully addressed the opportunities for improvement you highlighted.
>
>
> ---
> > Failure case analysis
>
> We sincerely appreciate your suggestions and will include an in-depth failure case analysis section in revision.
> We observe several consistent failure patterns, as shown in the  **[Figure](https://postimg.cc/18j0mVt4)**. For deeper analysis, we additionally prompted the VLMs to output their reasoning before answering. Below, we summarize the failure patterns in the order corresponding to the images.
> 1. Shape recognition can fail in VLMs, causing them to revert to priors rather than accurately interpreting visual input.
> 2. Models sometimes struggle to count accurately. Instead of performing an actual count, they default to relying on learned priors to estimate quantities.
> 3. Models may refuse to accept visual information that contradicts their learned priors, whereas humans can comprehend hypothetical scenarios. For instance, the model recognizes the city as Beijing but rejects the correct answer because it expects the Eiffel Tower to be in Paris.
> 4. For images with creative concepts, the model may overly rely on its learned priors. As illustrated, a common prior is that microscopes are used to view organisms, leading the model to answer “microscope” rather than identifying the creatively depicted stapler.
> 5. For images with blended features, the model may rely mostly on text input while overlooking the visual cues. As illustrated, the VLM heavily depends on textual input leading to saxophone as the answer.
>
> ---
> > Fine-grained evaluation and LLaVA-OneVision performance
>
> Thanks for the suggestions! Currently, we explicitly ask VLMs to produce a single-word response for evaluation, as it is  (1) cost-efficient without calling LLM, (2) fast, and (3) reliable. For robustness, we implemented a comprehensive synonym-matching pipeline to validate synonymous responses accurately. We indeed penalize responses that include long sentences, as we believe instruction-following is a foundation capability of VLMs—thus, failure to comply warrants a penalty.
>
> We acknowledge the value of your suggestion and have extended our evaluation system to include three categories: **Correct**, **Wrong**, and **Invalid**. “Invalid” refers to responses that do not meet the single-word requirement. Additionally, for “Invalid” outputs, we now provide an optional GPT-based evaluation that classifies them as either *Invalid-Correct* or *Invalid-Wrong*. The results is shown in **[Table](https://postimg.cc/SXgTQ1B4)**. The new evaluation enables us to assess the model’s performance which generates longer sentences.
>
>
>
> ---
> > Regarding Winoground and HallusionBench benchmarks and unrealistic examples
>
> For clarification, Appendix A.3 provides both quantitative and qualitative comparisons between our dataset and existing benchmarks such as WinoGround and HallusionBench, with key differences highlighted, ranging from high-level design principles to low-level formatting details.
>
> Moreover, our human study shows that participants can identify unnatural elements like tigers or moons. This underscores the human ability to reason about unrealistic or novel scenarios—an essential goal for advancing VLMs—and positions our dataset as complementary to existing benchmarks.
>
>
> ---
> > Reference of [1]
>
> Thanks for the suggestion, and we will discuss it [1] in the final revision. The key difference between ImageDPO and [1] is the training objective: while [1] follows the standard DPO setup by using preferred/dispreferred outputs for the same input, our approach fixes the output and varies the input conditions. This results in notable differences, like the distinct gradient behavior in Figure 13. Additionally, we introduce Proposition 1 and its theoretical proof in Appendix B, connecting our modified objective to the original DPO formulation.
>
> [1] Self-Supervised Visual Preference Alignment
>
>
> ---
> > Connection between Image-DPO and the benchmark
>
> Directly adjusting VLMs to improve specific abilities becomes challenging as the development of VLMs..  Our Image-DPO method addresses learned priors in a nuanced manner, where the proposed objective encourages the model to rely more heavily on the vision branch. Our objective drives a distinct shift in gradient directions when processing normal images $I_w$ compared to corrupted images $I_l$ (as illustrated in Figure 13). We appreciate your suggestion and will revise our wording to improve the connection.
>
> ---
> > Potential better approach
>
> We explored a similar idea but faced a key limitation: existing image-editing models often introduce visible artifacts or unnatural transitions between edited regions and the original background, making it easier for VLMs to detect anomalies. Additionally, they frequently struggle to follow instructions accurately. As editing technology improves, we plan to revisit this approach for future dataset extensions.

---

### Official Review · Reviewer_j6Xe · 2025-03-14

**Overall Recommendation:** 4

**Summary:**

This paper introduces ViLP, a benchmark to investigate how VLMs relies heavily on textual priors, and ignore visual inputs. ViLP consists of 300 questions (each including a distractor fact), and three image-answer pairs (a prior answer and two test answers). With extensive experiments, the authors show that existing VLMs struggle with this benchmark.
To improve the performance of VLMs, the authors also propose Image-DPO, a fine-tuning strategy (based on DPO approaches) that focus on the visual space/pixel space. The authors propose a method for generating training examples (seeded from COCO, Text2VQA, Visual Genome). Improvements of +2 to +3% are shown in their proposed benchmark.

**Claims And Evidence:**

- Claim 1: VLMs ignore visual inputs and are biased towards textual priors -- this is backed and supported by empirical results. Extensive experiments on their proposed benchmark (ViLP) support this claim.

- Claim 2: this can be fixes via Image-DPO: experimental results show some gains with their proposed method.

- Challenging dataset: the methodology for dataset creation is sound and backed by human/manual evaluation (Appendix A.2)

**Essential References Not Discussed:**

No essential references missing (to the best of my knowledge)

**Experimental Designs Or Analyses:**

Experimental analysis and design seems valid and sound.

**Methods And Evaluation Criteria:**

The method, while simple, shows improvement over the baseline. The only problem is the lack of evaluation of more recent models that might perform better in the proposed benchmark (e.g., QwenVL, DeepSeekVL, etc).

Image-DPO: leveraging existing datasets and modifying them via editing models/image generation approaches is time consuming, and the improvement seem marginal (~2 to 3%). However this is an interesting approach that future work could expand and build upon.

**Other Comments Or Suggestions:**

No additional comments

**Other Strengths And Weaknesses:**

The only weakness is the lack of comparison of their proposed Image-DPO to finetune more recent open VLMs (including LLaVA-NeXT, InstructBLIP, QwenVL and DeepSeekVL)

**Questions For Authors:**

Why is it necessary to finetune using additional synthetic images?

**Relation To Broader Scientific Literature:**

Prior works focus on benchmarking real-world data. VLMs tend to benefit from real-world common sense to answer questions in VQA datasets. The proposed benchmark revisits and shifts this particular limitation.

**Theoretical Claims:**

The proposed method seems correct, and aligns with the authors claims.

---

> ### Author Rebuttal · Authors · 2025-04-01
>
> We sincerely appreciate Reviewer j6Xe QTDS’s thoughtful and encouraging feedback. Due to constraints in time and computational resources, we have prioritized addressing your question and will incorporate the suggested additional improvements in future revisions.
>
> ---
> > Why is it necessary to fine-tune using additional synthetic images?
>
> This is a very good point! We employ a comprehensive pipeline that uses multiple image-generation models to synthesize additional images from seed datasets. Our primary motivation is diversity: by incorporating a broader range of synthetic images, the VLM is exposed to more varied content and can, in turn, produce more diverse Question–Image–Answer (QIA) pairs. This process also implicitly leverages the knowledge locked in the existing image-generation models. Compared to relying solely on the original dataset, these additional QIA pairs enrich the training signal, helping the model learn more robust and generalizable representations.
>
> To verify this point, we conducted an ablation study using the same dataset as in our ImageDPO experiments. Specifically, we directly generated QA pairs from images in the COCO, VG, and Text2VQA datasets without additional modifications or synthetic generation, i.e., no synthetic images. We then applied image corruptions to produce altered versions of these images and utilized these corrupted-image QA pairs for ImageDPO finetuning. The results, shown below, indicate notable performance drops. This outcome underscores that leveraging synthetic data, rather than relying solely on the original datasets, benefits the DPO training process.
>
> |                          | $ViLP^F_{Score}$ | $ViLP^F_{Prior}$ | $NB_Q$ | $NB_I$ | $NB_G$ | $NB_B$ | MM-Vet | $CHAIR^S \downarrow$ | $CHAIR^I \downarrow$ |
> | ------------------------ | ---------------- | ---------------- | ------ | ------ | ------ | ------ | ------ | -------------------------------------- | -------------------------------------- |
> | Ours                     | 34.17 | 39.33 | 39.79 | 45.47 | 14.16 | 68.45 | 32.3 | 45 | 12.3 |
> | Without synthetic images | 31.17 | 38 | 38.32 | 44.17 | 13.35 | 67.69 | 31.4 | 47 | 13.6 |
>
>
> ---
> > The only weakness is the lack of comparison of their proposed Image-DPO to finetune more recent open VLMs (including LLaVA-NeXT, InstructBLIP, QwenVL and DeepSeekVL)
>
> Thank you for this valuable suggestion! We agree that comparing Image-DPO with more recent VLMs (e.g., LLaVA-NeXT, InstructBLIP, QwenVL, DeepSeekVL) would provide additional insights. However, due to limited time and computational resources during the rebuttal period, we are unable to conduct these comparisons comprehensively. We appreciate this feedback and plan to include parts of the models in our future revisions. Nonetheless, we believe our current results on LLaVA-v1.5 and Cambrian sufficiently demonstrate Image-DPO’s effectiveness and hope they convey the potential of our approach.

---

### Official Review · Reviewer_1kEP · 2025-03-21

**Overall Recommendation:** 4

**Summary:**

The paper presents an investigation into the over-reliance of vision-language priors in existing Vision-Language Models (VLMs). To enable it, the paper presents ViLP, a carefully designed benchmark which consists of Prior Answer that can be directly inferred from the question, and Test Answers which rely on visual cues. Their evaluation reveals that all methods significantly fall short of human annotations. To improve upon this shortcoming, ImageDPO is developed, the core of which is to develop pair-wise images where a corrupted image is generated from a real image using a VLM-based pipeline which uses techniques and models such as image editing and Grounded SAM. The authors theoretically prove ImageDPO and empirically show that when applied to OSS models such as LLaVA, ImageDPO improves upon other methods in solving ViLP.

**Claims And Evidence:**

1. Overall, I like the paper structure. The paper introduces a new benchmark, has interesting experiments and ablations, and a DPO-based methods that help improve the benchmark.
2. The ViLP dataset is well motivated -- humans do well despite having distractors whereas VLM's clearly fail as shown by the paper's evaluation protocol.
3.  ImageDPO is intuitive as well, using divergent image data to force the model to learn through image differences. The results are promising as well.

**Essential References Not Discussed:**

N/A

**Experimental Designs Or Analyses:**

I checked the soundness of the experiments. I am convinced, overall.

**Methods And Evaluation Criteria:**

1. Overall, I am convinced with the methodology adopted in the paper.
2. I have the following question :
a) Since data generation for DPO training uses image-editing, I wonder if there is some quality control established to threshold what is an accepted corrupted image?

**Other Comments Or Suggestions:**

N/A

**Other Strengths And Weaknesses:**

1. The paper could do with a bit of re-writing and typos. For example, Line 1152.

**Questions For Authors:**

Please check above.

**Relation To Broader Scientific Literature:**

The paper covers previous related work and evaluates a large range of models, both open and closed-sourced.

**Theoretical Claims:**

1. The paper derives the ImageDPO in Appendix B. I do not find issues with it.

---

> ### Author Rebuttal · Authors · 2025-04-01
>
> We sincerely appreciate the positive review provided by  Reviewer 1kEP, and are grateful that Reviewer 1kEP “likes the paper structure”, and “convinced with the methodology adopted in the paper”,  believes “the  ViLP dataset is well motivated”, and “the results are promising as well”.
> Below, we provide our detailed responses to your comments.
>
> ---
>
> > I have the following question: a) Since data generation for DPO training uses image-editing, I wonder if there is some quality control established to threshold what is an accepted corrupted image?
>
> That's a good point! Our guiding principle is to eliminate human involvement in order to make the pipeline fully automatic and scalable. While the generated data may contain some noise, we believe the pipeline is robust enough to help improve VLM performance, which is supported by the improvements in the experiments. For corrupted images, we control the hyperparameters of image editing, such as the strength of Gaussian blur and pixelation, to ensure that the corrupted versions are noticeably worse than the original ones. In the case of semantic corruption, we calculate the number of pixels that are modified and discard cases where the changes affect less than 5% of the total pixels, as such minor alterations may be too subtle to detect.
>
> ---
>
> > The paper could do with a bit of re-writing and typos. For example, Line 1152.
>
> Thank you for your thorough review! We have corrected the typo on line 1152 in the latest revision and will carefully proofread and refine the manuscript to enhance readability further.

---

> > ### Comment · Reviewer_1kEP · 2025-04-05
> >
> > I am satisfied with the rebuttal and have no further questions. I keep my score.

---

### Official Review · Reviewer_4vBy · 2025-03-23

**Overall Recommendation:** 4

**Summary:**

This paper investigated the problem of over-reliance on visual language priors of Vision-Language Models (VLMs) instead of visual reasoning. To study this problem, this work further proposed a benchmark ViLP containing 900 question-image-answer triplets, covering both prior answers and test answers, revealing that GPT-4o only scored at ~66%. This work further proposed Image-DPO, a
self-improving approach to enhance VLM visual reasoning performance by increasing reliance on visual inputs, applied on LLaVA-v1.5 and Cambrian.

**Claims And Evidence:**

The claims are supported by clear and convincing evidence.

**Essential References Not Discussed:**

No.

**Experimental Designs Or Analyses:**

The overall soundness/validity of the experiments look good.

**Methods And Evaluation Criteria:**

The proposed methods and benchmark dataset make sense.

**Other Comments Or Suggestions:**

Please see Weaknesses

**Other Strengths And Weaknesses:**

Strengths:

* The paper is well-written and organized. The three criterions are clear and mathematical to understand how the vision-language priors are scoped.

* The Image-DPO solution makes sense and is aligned with the motivation to highlight visual clues. Table 3 and 4 show that Image-DPO outperforms baselines and other preference optimization methods.

Weaknesses & Questions:

* The images are generated by DALL·E-3 and Flux, which look quite different from real-world images. The cartoon style may have impacts on VLMs as they are not directly trained on synthetic images. If real-world but rare images or realistic-style (or more styles) can be included to ablate the impacts of image styles. Therefore, the conclusions of this paper should be bounded by cartoon style.

* It is not clear about the role of Proposition 1 as it is not mentioned in the main text after being introduced.

* Some answers to the counterfactual questions are ambiguous. Take Figure 3 as examples, for the upper right example, it is hard to tell the cities are London and Paris, as Moscow can also build rivers or Tour Eiffel model. Also for the lower right example, the camel can also have a horse face by wearing a mask. There lacks strict definitions of these nouns and it is hard to tell the MLLMs' answers are not reasonable. The reasons might be that the models are forced to output a single word as answers. Actually, the explanations of the answer can provide more information on how the model observe the image.

**Questions For Authors:**

Please see Weaknesses. My main concerns are the style of images and the ambiguous answers due to the single-word evaluation setting. My current score is actually borderline.


--------------

After reading the rebuttal, I decided to increase the score to accept. The analysis on the impact of image style is insightful and answered my questions. Since my major concern is fully addressed, I am happy to accept the paper.

**Relation To Broader Scientific Literature:**

No.

**Theoretical Claims:**

The paper does not include theoretical claims.

---

> ### Author Rebuttal · Authors · 2025-04-01
>
> We greatly appreciate the constructive feedback from Reviewer 4vBy. We seriously address your comments below.
>
> ---
>
> > Regarding the style of images.
>
> We appreciate this valuable point and acknowledge that some generated images appear cartoon-like or synthetic, due to the image generation tools selected—among the best available at the time. However, since our objective is to benchmark VLMs using out-of-distribution QIAs that are not readily available online, we adopt image generation models to achieve this goal – a motivation appreciated by Reviewer j6Xe *“Prior works focus on benchmarking real-world data…The proposed benchmark revisits and shifts this particular limitation”*.
>
> Regarding the cartoon-style concern, we would like to respectfully point out that current VLMs are commonly trained on datasets that include synthetic and cartoon-style images. For example, [LAION](https://laion.ai/blog/laion-aesthetics/), widely used in training VLMs, contains subsets focused on *anime*, *cartoon*, and *aesthetics* (filtered by community), reflecting the prevalence of synthetic imagery online. VLMs have shown robust performance on such data, and understanding synthetic visual representations remains an important capability, especially given humans’ ease in interpreting them.
>
> To assess the impact of image style, we regenerated a subset of 45 QIA pairs using GPT-4o’s latest image generation model to enhance realism (**[Figure](https://postimg.cc/dZTtNwtW)**). We then measured changes in model correctness when using these realistic images, with negative values indicating performance degradation. The results show that increased realism slightly reduces performance in most cases for the “Score” metric, while most effects on “Prior” are zero. These ablation results suggest that incorporating more realistic images may increase task difficulty, highlighting an important direction for future research.
>
> | | $ViLP^F_{Prior}$ | $ViLP^F_{Score}$ | $ViLP^P_{Prior}$ | $ViLP^P_{Score}$ |
> | -| - | - | -| -|
> | GPT-4o                                  | 0 | -2.2% | 0 | 0 |
> | GPT-4o-mini                                    | 0                | -1.1%               | 0                | -2.2%               |
> | Claude-Sonnet-3.5                              | -2.2%               | -4.4%               | 0                | -3.3%               |
> | Claude-Opus-3                                  | 2.2%                | -1.1%               | -2.2%               | -1.1%               |
>
>
> ---
>
>
> > Regarding Proposition 1.
>
> We would like to improve this point in the revision. For clarification, proposition 1 is the theoretical foundation of our Image DPO algorithm, showing that minimizing our proposed Image DPO loss corresponds to minimizing an upper bound on the RLHF loss. Unlike typical DPO modifications that adjust the output, our method fixes the output and alters the conditioning. Proposition 1 thus rigorously establishes the connection between our modified objective and the original DPO formulation.
>
>
> ---
>
> > Regarding the ambiguity and single-word evaluation.
>
> We thank the reviewer for highlighting the inherent ambiguity in counterfactual visual reasoning. Although ambiguous scenarios occur, humans typically resolve them effectively after initial confusion, as evidenced by the strong human performance reported in our benchmark (Table 2), where evaluators had access to identical textual and visual inputs as the VLMs. Thus, the noted "ambiguity" intentionally examines whether VLMs can match human performance under controlled conditions.
>
> Regarding single-word outputs, we highlight their evaluation advantages—including efficiency, speed, and reliability—when combined with our synonym and plural detection pipeline. In contrast, sentence-based evaluations relying on LLM inference incur API fees, computational costs, sensitivity to hyperparameters, and occasional unreliability due to incorrect reasoning.
>
> We appreciate the reviewer's concerns about potential limitations of single-word answers. To explore this, we conducted ablation studies prompting GPT-4o in a "chain-of-thought" manner, instructing it to reason before giving a sentence-level final answer. We evaluated these answers against ground truths using GPT, comparing both reasoning steps and final sentences with original single-word responses.
>
> Interestingly, allowing longer answers with more reasoning tokens did not yield performance benefits. As shown in (**[Figure](https://postimg.cc/XrpqpR57)**),  correct reasoning (second pic.) can still produce incorrect conclusions, and sometimes the reasoning process itself may be flawed (third pic.). Quantitatively, GPT-4o’s performance on "Prior" slightly improved from 91% to 92.67%, whereas on "Score" it decreased from 66.17% to 55.5%. Given that "Prior" aligns closely with language priors and "Score" represents out-of-distribution scenarios, we hypothesize that longer outputs may unintentionally increase reliance on priors during reasoning.

---

### Decision · Program_Chairs · 2025-05-01

**Decision:**

Accept (poster)

**Comment:**

The paper introduces a benchmark, called ViLP, which constructs OOD image-text pairs to evaluate the reliance of VLMs on the language prior acquired during training to answer visual questions. The dataset is relatively small (contains only 300 examples) but the findings remain quite interesting and valuable to the community. For instance, the authors find that while humans can answer the questions easily, VLMs don’t (e.g. GPT4 achieves 66% accuracy). In addition, the authors recommend a DPO-like optimization algorithm for VLMs that is simple to implement and seems to outperform self-improvement baselines (which I find to be surprising). Proposition 1 helps give some insight for why the method works. Overall, this is a nice, insightful paper that is well-written. Overall, all reviewers and I have found the paper to be a valuable contribution.